# RYBP stimulates PRC1 to shape chromatin-based communication between Polycomb repressive complexes

Nathan R Rose[1†], Hamish W King[1†], Neil P Blackledge[1†], Nadezda A Fursova[1], Katherine JI Ember[1], Roman Fischer[2], Benedikt M Kessler[2,3], Robert J Klose[1,3*]

[1]Department of Biochemistry, University of Oxford, Oxford, United Kingdom; [2]TDI Mass Spectrometry Laboratory, Target Discovery Institute, University of Oxford, Oxford, United Kingdom; [3]Nuffield Department of Medicine, University of Oxford, Oxford, United Kingdom

**Abstract** Polycomb group (PcG) proteins function as chromatin-based transcriptional repressors that are essential for normal gene regulation during development. However, how these systems function to achieve transcriptional regulation remains very poorly understood. Here, we discover that the histone H2AK119 E3 ubiquitin ligase activity of Polycomb repressive complex 1 (PRC1) is defined by the composition of its catalytic subunits and is highly regulated by RYBP/YAF2-dependent stimulation. In mouse embryonic stem cells, RYBP plays a central role in shaping H2AK119 mono-ubiquitylation at PcG targets and underpins an activity-based communication between PRC1 and Polycomb repressive complex 2 (PRC2) which is required for normal histone H3 lysine 27 trimethylation (H3K27me3). Without normal histone modification-dependent communication between PRC1 and PRC2, repressive Polycomb chromatin domains can erode, rendering target genes susceptible to inappropriate gene expression signals. This suggests that activity-based communication and histone modification-dependent thresholds create a localized form of epigenetic memory required for normal PcG chromatin domain function in gene regulation.

*For correspondence: rob.klose@
bioch.ox.ac.uk

†These authors contributed
equally to this work

Competing interests: The
authors declare that no
competing interests exist.

Reviewing editor: Peter
Verrijzer, Erasmus University
Medical Center, Netherlands

## Introduction

In multicellular eukaryotes the capacity to express defined subsets of genes at the correct time and place underpins normal development. While genetically encoded regulatory pathways define the nature of these gene expression patterns in individual cell types and tissues, these systems must ultimately converge on a DNA substrate that is not unadorned, but instead wrapped around histone proteins to form nucleosomes and chromatin. Through studying chromatin function in the context of gene regulatory networks it is now clear that modification of DNA and histones is essential for controlling normal regulation of gene expression (*Bannister and Kouzarides, 2011*).

In animals, Polycomb group (PcG) proteins play central roles in gene regulation through their chromatin modifying activities. First identified in *Drosophila* as important regulators of body plan specification, orthologous PcG systems were subsequently identified in vertebrates and also shown to be essential for developmental gene regulation (*Lewis, 1978*; *Schwartz and Pirrotta, 2013*). Biochemical characterization of PcG proteins has revealed that they form large multiprotein complexes that function as chromatin-modifying or -remodelling enzymes (*Müller and Verrijzer, 2009*; *Simon and Kingston, 2013*). PcG proteins generally fall into one of two multiprotein complexes, called Polycomb repressive complex 1 and 2 (PRC1 and PRC2). PRC1 functions as a histone E3 ubiquitin ligase that mono-ubiquitylates histone H2A at position 119 (H2AK119ub1) (*Cao et al., 2005*; *de Napoles et al., 2004*; *Wang et al., 2004a*) and PRC2 is a histone H3 lysine 27 (H3K27)

methyltransferase (*Cao et al., 2002*; *Czermin et al., 2002*; *Müller et al., 2002*; *Margueron and Reinberg, 2011*). Together, the activities of these two PcG protein complexes play important roles in maintaining repressive chromatin states and controlling gene expression during cell fate transitions (*Xie et al., 2013*; *Arnold et al., 2013*; *Boyer et al., 2006*; *Leeb and Wutz, 2007*; *Endoh et al., 2008*; *Terranova et al., 2008*; *Koche et al., 2008*; *Bracken et al., 2006*).

The defined molecular mechanisms that underpin how PcG complexes regulate transcription still remain poorly understood. However, it has been proposed that the occupancy of PRC1 and PRC2 at genomic target sites and their capacity to modify histones may function to create structurally compacted chromatin which is inhibitory to transcription (*Eskeland et al., 2010*; *Francis et al., 2004*; *Wani et al., 2016*; *Boettiger et al., 2016*). To achieve this, the function of PRC1 and PRC2 appear to be intimately linked. For example, some PRC1 complexes are able to recognise and bind PRC2-dependent H3K27me3, recruiting those PRC1 complexes to sites occupied by PRC2 (*Min et al., 2003*; *Wang et al., 2004b*). Similarly, PRC2 can recognize PRC1-deposited H2AK119ub1, regulating PRC2 occupancy on chromatin (*Blackledge et al., 2014*; *Cooper et al., 2014*). Furthermore, in vitro assays have suggested that H2AK119ub1 may also stimulate the enzymatic activity of PRC2 (*Kalb et al., 2014*). Based on these observations it has been proposed that feedback mechanisms, whereby one Polycomb repressive complex supports the recruitment and possibly activity of the other complex, may in part underpin chromatin-based transcriptional repression (*Blackledge et al., 2015*; *Steffen and Ringrose, 2014*; *Voigt et al., 2013*). In other scenarios the relevance of histone modifications to gene repression, particularly the role of H2AK119ub1, has remained unclear, with the repressive nature of PRC1 being proposed to reside within its capacity to alter chromatin structure and compaction independently of its H2A ubiquitylating activity (*Pengelly et al., 2015*; *Illingworth et al., 2015*; *Francis et al., 2004*; *Eskeland et al., 2010*). However in most cases, the defined contribution of H2AK119ub1-dependent and –independent pathways, and their relevance to Polycomb chromatin domain function in vivo have remained incompletely defined.

In order to understand the mechanisms by which the PcG systems function together in vivo, it will be critical to discover how, if at all, their enzymatic activities contribute to chromatin structure and gene regulation. While significant effort has been placed on understanding in vitro how the H3K27 methyltransferase activity of PRC2 is controlled (*Margueron et al., 2009*; *Jiao and Liu, 2015*; *Cao and Zhang, 2004*), far less is known about how PRC1 complexes catalyse H2AK119ub1 and how this relates to the function of PRC1 in vivo. This paucity in our understanding of PRC1 activity is in part due to the diverse nature of individual PRC1 complexes. At the simplest level, PRC1 complexes are formed by their catalytic subunit, RING1A or RING1B, which dimerises with a PCGF protein to form an active E3 ubiquitin ligase that targets histone H2A (*Levine et al., 2002*; *Satijn et al., 1997*). In mammals, there are six PCGF proteins (PCGF1–6), and the identity of the PCGF protein incorporated into PRC1 appears to direct interaction with a specific complement of auxiliary proteins, giving rise to an array of diverse PRC1 complexes (*Blackledge et al., 2014*; *Gao et al., 2012*; *Tavares et al., 2012*). Based on the protein subunit composition of these individual PRC1 complexes, they are often then divided into 'canonical' or 'variant' complexes. Canonical PRC1 complexes form around either PCGF2 or 4, and incorporate a chromobox (CBX) protein, which can directly bind to PRC2-catalysed H3K27me3 (*Fischle et al., 2003*; *Bernstein et al., 2006*; *Min et al., 2003*). In contrast, variant PRC1 complexes can form around all six PCGF proteins, but these complexes engage with RYBP or YAF2, two proteins that bind to a site on RING1B that is mutually exclusive with CBX protein binding (*García et al., 1999*; *Wang et al., 2010*).

Despite the diversity of PRC1 complexes, biochemical characterization of their catalytic activity has largely been focused on canonical PCGF2/PCGF4-containing complexes (*Li et al., 2006*; *Gao et al., 2012*; *Elderkin et al., 2007*; *Cao et al., 2005*; *Buchwald et al., 2006*; *Bensaadon et al., 2006*). For example, in vitro ubiquitylation assays using reconstituted complexes and nucleosome substrates demonstrated that RING1(A/B) could form an active E3 ligase in the presence of either PCGF2 or PCGF4 (*Elderkin et al., 2007*; *Buchwald et al., 2006*). Furthermore, a minimal complex comprised of a dimer between the N-terminal RING finger domains of PCGF4 and RING1B was sufficient to deposit H2AK119ub1 (*Buchwald et al., 2006*). Importantly, the RING finger domains of individual PCGF proteins are highly similar, suggesting that they will likely all support formation of an active E3 ligase complex. However, more recent work examining PCGF(1–6)/RING1(A/B) catalytic RING-RING dimers revealed that PCGF(2/4)/RING1(A/B) were less active than PCGF

(1/3/5/6)/RING1(A/B) in autoubiquitylation and E2 lysine discharge assays (*Taherbhoy et al., 2015*). Surprisingly, when these same assemblies were challenged with physiologically relevant nucleosome substrates, complex-specific activities were no longer apparent. Nevertheless, consistent with the possibility that individual PRC1 complexes may have distinct activities related to protein complex composition, in vivo studies in both mammals and *Drosophila* have suggested that less-well characterized variant PRC1 complexes may contribute more significantly to H2A mono-ubiquitylation than canonical PRC1 (*Tavares et al., 2012*; *Lee et al., 2015*; *Lagarou et al., 2008*; *Blackledge et al., 2014*). For example, the *Drosophila* dRAF complex, which appears to have similarities to mammalian variant PRC1 complexes, is a highly active H2A ubiquitin ligase in vitro and appears to be the predominant H2A ubiquitin ligase in *Drosophila* S2 cells (*Lagarou et al., 2008*). Furthermore, a PCGF3 orthologue contributes widely to H2AK119ub1 in cultured *Drosophila* cells (*Lee et al., 2015*). This raises the possibility that individual PCGF proteins might confer differing E3 ligase activities on PRC1, either directly, or via interaction with complex-specific regulatory subunits. Indeed, incorporation of the auxiliary subunits RYBP, PHC2 or CBX2/8 into a PCGF4/RING1B complex has been reported to affect its E3 ligase activity in different ways (*Gao et al., 2012*), though this was not the case for PCGF2/RING1B, in which auxiliary subunits had little impact on catalytic activity (*Tavares et al., 2012*). Therefore a more detailed understanding of how H2AK119 ubiquitin ligase activity is shaped by different PRC1 assemblies, both in vitro and in vivo, is required to better understand PRC1 involvement in polycomb chromatin domain function and gene regulation.

To address these important questions we have exploited biochemical and genomic approaches to interrogate PRC1 enzymatic activity in vitro and in cells. This revealed that PRC1 complexes are highly modular, and that the enzymatic activity of PRC1 is defined by the choice of PCGF protein combined with the influence of RYBP/YAF2, uncovering an enzymatic logic within the assembly of individual PRC1 complexes. Building on these discoveries, genetic ablation and genomic approaches uncover a widespread role for RYBP in stimulating H2AK119 E3 ligase activity in vivo, and we demonstrate that the absence of this activity leads to defects in the deposition of H3K27me3 by PRC2. Importantly, in the absence of RYBP, loss of PRC1 stimulation and subsequent defects in PRC2 activity can lead to erosion of Polycomb chromatin domains and susceptibility to inappropriate gene activation signals. Together, these observations identify an activity-based communication between PRC1 and PRC2 in vivo, and suggest that histone modification-dependent thresholding mechanisms sustain Polycomb chromatin domains, connecting the activity of PcG systems to their occupancy on chromatin and transcriptional repression.

## Results

### PCGF1-PRC1 is a highly modular protein complex

In embryonic cells and tissues where the PcG system is thought to play a pivotal role in developmental gene regulation, the PCGF1-PRC1 complex is highly abundant (*Kloet et al., 2016*), and is required for H2AK119ub1 and normal development (*Farcas et al., 2012*; *Blackledge et al., 2014*; *Gao et al., 2012*; *He et al., 2013*; *Wu et al., 2013*; *Gearhart et al., 2006*; *Boulard et al., 2015*). Furthermore, tethering experiments had previously suggested that the variant PCGF1 complex might be more active than PCGF2/4 complexes in vivo (*Blackledge et al., 2014*). Given the essential nature of this PRC1 complex and its apparent elevated activity in vivo, we sought to use it as a model system to understand the biochemical mechanisms that underpin the assembly and catalytic activity of PRC1. To achieve this, we used a baculovirus co-expression system (*Bieniossek et al., 2012*) to assemble and affinity-purify the five component PCGF1-PRC1 holocomplex containing RING1B, PCGF1, RYBP, KDM2B and BCOR (*Figure 1A and B*). To understand in more detail the topology of the PCGF1-PRC1 complex we subjected it to crosslinking followed by tandem mass spectrometry analysis, which identifies the relative proximity of proteins within large protein complex assemblies (*Leitner et al., 2014*). This revealed clear spatial relationships between individual components of the complex (*Figure 1C*). For example, RING1B was in close proximity to PCGF1 and RYBP. Importantly RING1B and RYBP crosslinks were consistent with atomic resolution information detailing their physical interaction (*Figure 1—figure supplement 1*) (*Wang et al., 2010*). BCOR and KDM2B crosslinked extensively with each other suggesting they are in close proximity and also displayed additional crosslinks with PCGF1 and RYBP. However, there was no evidence for significant

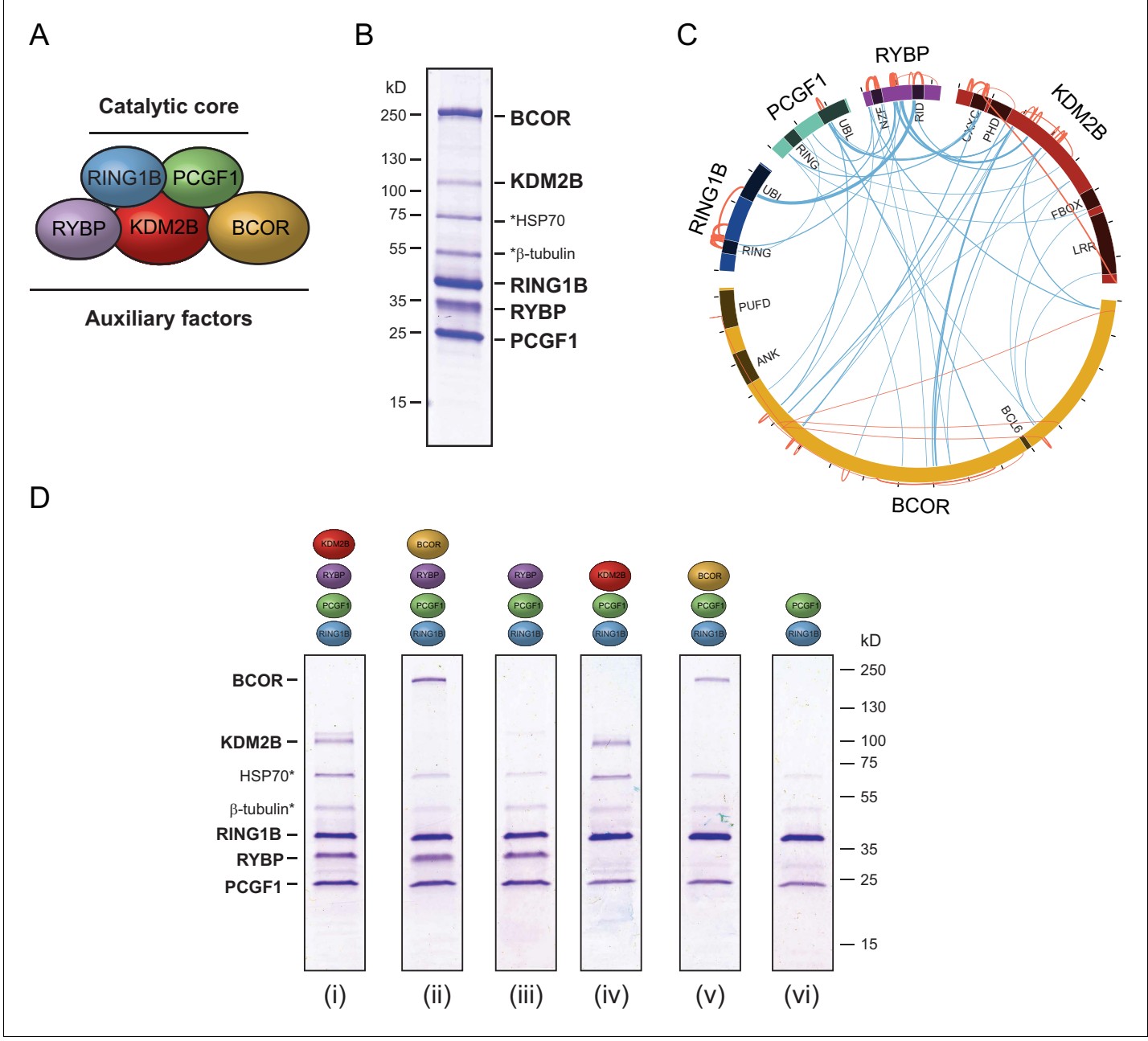

**Figure 1.** PCGF1-PRC1 is a highly modular complex. (**A**) A schematic of the PCGF1-PRC1 complex indicating the core catalytic dimer of RING1B and PCGF1 and auxiliary components RYBP, KDM2B, and BCOR. (**B**) A Coomassie-stained gel of PCGF1-PRC1 affinity-purified via RING1B. Individual subunits are labelled and * denotes contaminants HSP70 and β-tubulin. (**C**) A Circos plot illustrating crosslinking mass spectrometry analysis of the PCGF1-PRC1 complex reveals extensive interactions between PCGF1-PRC1 subunits. Blue and orange lines indicate intermolecular and intramolecular crosslinks respectively. Lines are weighted by the statistical confidence of their interaction. (**D**) Coomassie-stained gels of affinity-purified PCGF1-PRC1 sub-complexes demonstrating the modularity of PCGF1-PRC1 complex. Complexes were affinity-purified via RING1B. Above each lane (i-vi) the co-expressed factors are illustrated and * denotes contaminants HSP70 and β-tubulin.

The following figure supplement is available for figure 1:

**Figure supplement 1.** Crystal structure validation of crosslinking mass spectrometry-identified proximities.

crosslinks between either BCOR or KDM2B and RING1B, suggesting they are not in close proximity within the complex. This suggests a high degree of modularity within the PCGF1-PRC1 complex, where individual combinations of proteins engage extensively with each other, while others exhibit little or no clear proximity.

Based on the apparent modularity of PCGF1-PRC1, we set out to dissect the requirement for individual components in its assembly. To achieve this we generated a series of co-expression vectors in which individual proteins were iteratively omitted (*Figure 1D*). This revealed that KDM2B and BCOR, which appear to interact with the complex through PCGF1 (*Figure 1C*), can be completely removed, either individually or in combination, to leave an intact PCGF1-RING1B-RYBP sub-complex (*Figure 1D(i–iii)*). Similarly, when RYBP was removed from the PCGF1-PRC1 complex, BCOR and KDM2B were still able to independently interact with the PCGF1-RING1B dimer (*Figure 1D(iv–v)*). This indicates that inclusion of BCOR and KDM2B in the PCGF1-PRC1 complex does not require RYBP, or an interaction with each other. Finally, in agreement with analysis of other PCGF/RING1 interactions, PCGF1/RING1B formed dimers in the absence of other PCGF1-PRC1 components (*Figure 1D(vi)*). Therefore, the PCGF1-PRC1 holocomplex assembles around a core PCGF1-RING1B dimer, and does so in a highly modular fashion. Interestingly, in *Drosophila*, the related dRAF complex exhibits similar modularity (*Lagarou et al., 2008*). This contrasts the assembly of the PRC2 holocomplex, which appears to rely on extensive and highly integrated protein-protein interactions for its efficient formation (*Ciferri et al., 2012*; *Jiao and Liu, 2015*).

## RYBP/YAF2 stimulates H2A ubiquitylation by PCGF1-PRC1

Within multiprotein enzyme complexes, non-catalytic components can function to regulate enzymatic activity, substrate recognition, or targeting. Our understanding of how PRC1 complexes function in these regards remains very poorly defined. However, the inherent modularity of the PCGF1-PRC1 complex provided a unique opportunity to examine if individual components of the complex contribute to the H2AK119 E3 ligase activity of PRC1. To address this important question, we first established a quantitative in vitro E3 ligase activity assay using nucleosome substrates and confirmed that reconstituted PCGF1-PRC1 was active by detecting the conversion of H2A to mono-ubiquitylated H2A (*Figure 2A*). We then compared the relative E3 ligase activity of the PCGF1-RING1B core dimer to the five component PCGF1-PRC1 holocomplex (*Figure 2B*). Surprisingly, the PCGF1-PRC1 holocomplex was substantially more active than the PCGF1-RING1B core (*Figure 2B*). To understand the molecular determinants controlling this activity, we then iteratively measured the H2AK119 E3 ligase activities of individual PCGF1-PRC1 sub-complexes (*Figure 2C*). This revealed that interaction of RYBP with the RING1B-PCGF1 core dimer was responsible for elevated enzymatic activity of the PCGF1-PRC1 holocomplex. In contrast, KDM2B and BCOR appeared to contribute little or nothing to this activity, suggesting that they confer alternative, non-enzymatic functions to the PRC1 complex. Variant PRC1 complexes can contain either RYBP or YAF2, and their inclusion appears to be mutually exclusive, presumably because they bind to the same region of RING1B (*Gao et al., 2012*; *Wang et al., 2010*; *Tavares et al., 2012*). Given the stimulatory activity of RYBP on PRC1, we wanted to determine whether YAF2 was also capable of stimulating PCGF1-RING1B. Therefore, we purified recombinant YAF2 and RYBP and added them to PCGF1-RINGB E3 ligase reactions (*Figure 2—figure supplement 1A*). Importantly, this revealed that YAF2 also stimulated PCFG1-RING1B activity (*Figure 2—figure supplement 1B*). Together these observations reveal that RYBP and YAF2 regulate PCGF1-PRC1 activity in vitro, where they act to stimulate the PCGF1-RING1B core dimer and enhance H2AK119ub1 deposition.

## The activity of PRC1 complexes is defined by their PCGF component and RYBP-dependent stimulation

Ectopic targeting of PRC1 complexes to chromatin in vivo had suggested that PCGF1-PRC1 is more efficient at depositing H2AK119ub1 on chromatin than PCGF4-PRC1 (*Blackledge et al., 2014*). Therefore, we sought to understand whether differences in E3 ligase activity between individual PRC1 complexes are inherent to the PCGF-RING1B catalytic core, or whether RYBP may have selective stimulatory activity. This was important as previous in vitro work had only directly compared the H2AK119ub1 E3 ligase activity of differing PCGF/RING1 assemblies in the context of minimal RING-RING catalytic dimers (*Taherbhoy et al., 2015*). Furthermore, when intact PCGF(2/4)/RING1 dimers

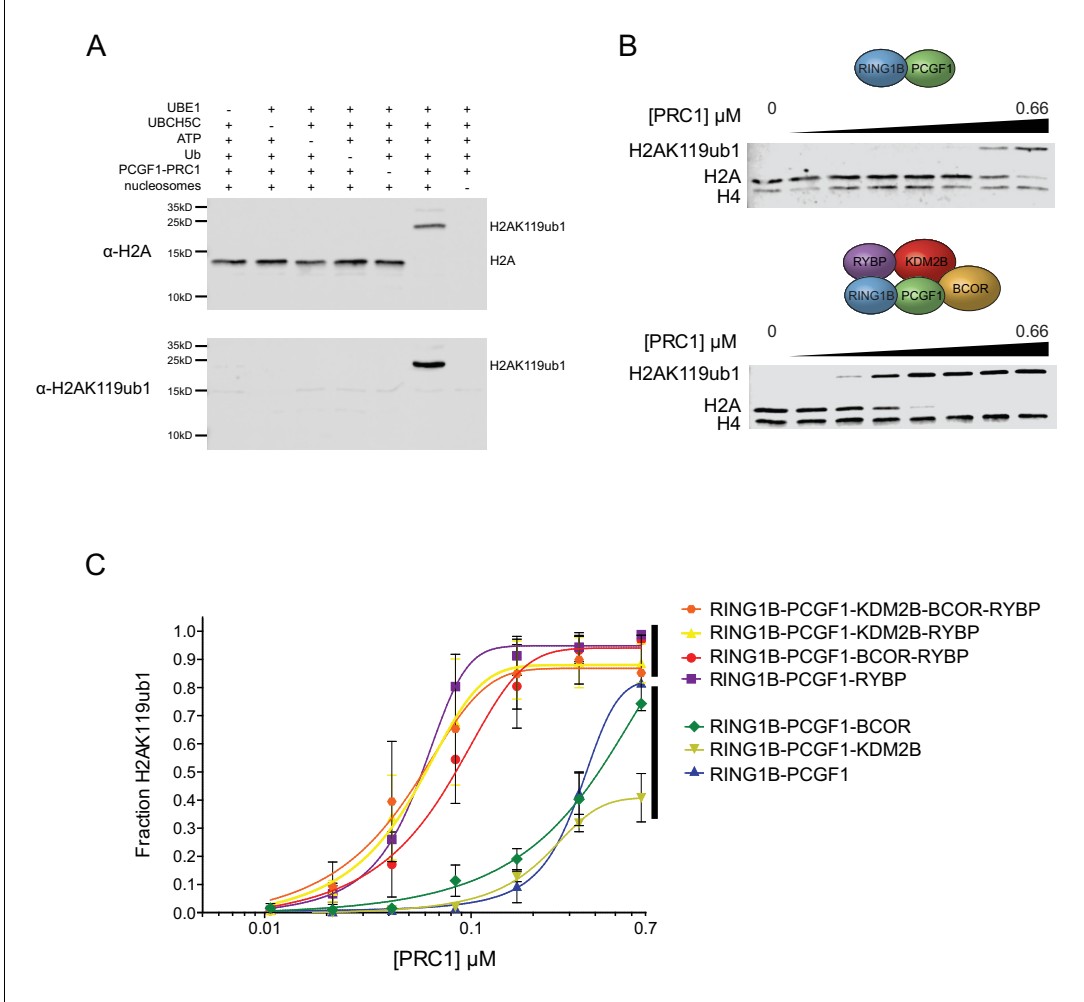

**Figure 2.** RYBP stimulates the catalytic core of PCGF1-PRC1. (**A**) A histone H2AK119 E3 ligase assay monitoring the conversion of H2A to mono-ubiquitylated H2A by western blotting, which demonstrates that the activity of PCGF1-PRC1 requires E1 (UBE1), E2 (UBCH5C), Ubiquitin and ATP. The top panel monitors H2A mono-ubiquitylation with an antibody against total H2A that detects both mono-ubiquitylated and non-ubiquitylated H2A. The bottom panel monitors production of mono-ubiquitylated H2A with an antibody that specifically recognizes H2AK119ub1. (**B**) An E3 ligase activity assay directly comparing the PCGF1-RING1B catalytic core and the PCGF1-PRC1 holocomplex over increasing enzyme concentrations (two fold dilution series) as examined by western blotting. (**C**) An E3 ligase activity assay comparing the PCGF1-PRC1 holocomplex and various sub-complexes in which individual and combinations of PCGF1-PRC1 subunits have been omitted (see *Figure 1D*). This demonstrates that RYBP is the central determinant of E3 ligase activity for the PCGF1-PRC1 complex. The error bars correspond to the S.E.M. of at least two independent experiments.

The following figure supplement is available for figure 2:

**Figure supplement 1.** YAF2 stimulates PCGF1-RING1B E3 ligase activity.

had been examined, differing conclusions were made as to whether RYBP stimulated E3 ligase activity (*Tavares et al., 2012*; *Gao et al., 2012*). Therefore to examine these points in more detail, we expressed and purified intact PCGF1-RING1B and PCGF4-RING1B dimers and compared their relative activities in H2AK119 E3 ligase assays (*Figure 3A*). Interestingly, this revealed that PCGF1-RING1B was significantly more active than PCGF4-RING1B (*Figure 3B*), indicating that the enzymatic activity of a given PRC1 complex is inherently defined by the PCGF-RING dimer which forms its catalytic core. Given that PCGF1-RING1B is robustly stimulated by RYBP, we were then curious to understand whether RYBP stimulated PCGF4-RING1B. This was an important question as previous studies examining the effect of RYBP on PCGF4-RING1B and the related PCGF2-RING1B dimer had come to differing conclusions, with one study observing no effect on activity and the other claiming

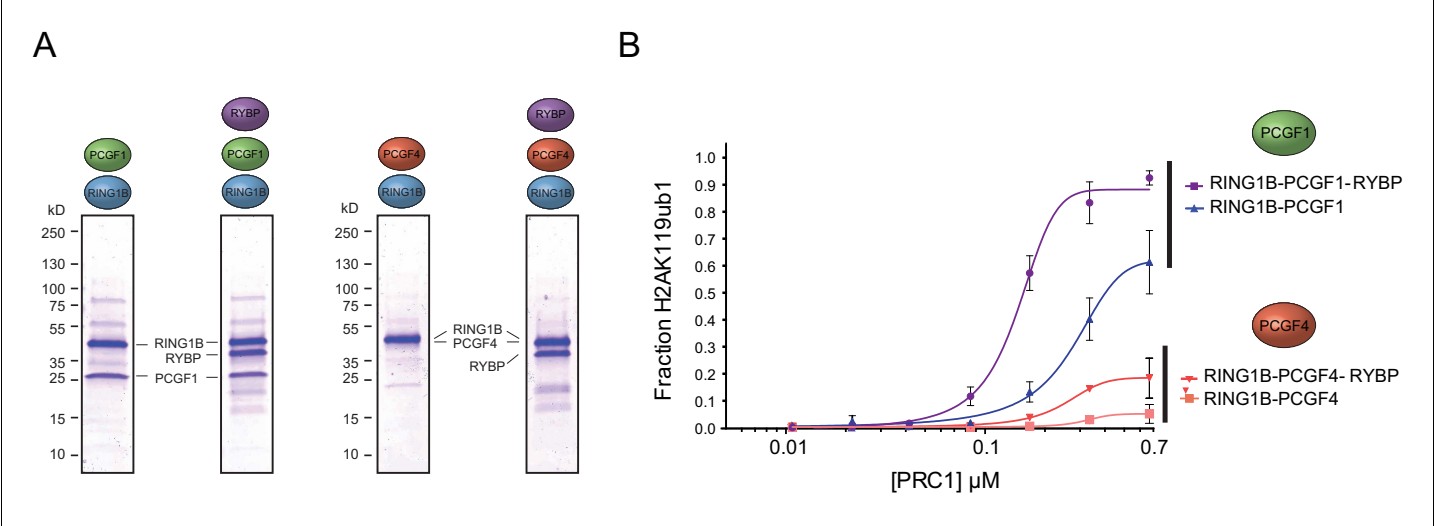

**Figure 3.** The PCGF1 catalytic dimer is inherently more active and stimulated more robustly than the PCGF4 catalytic dimer. (**A**) Coomassie-stained gels showing the recombinant PCGF1-RING1B or PCGF4-RING1B catalytic cores with or without the addition of RYBP. Complexes were affinity-purified via RING1B. PCGF4 and RING1B migrate at the same position on the gel (*Figure 3—figure supplement 1*). (**B**) An E3 ligase activity assay comparing PCGF1-RING1B or PCGF4-RING1B in the presence or absence of RYBP. The error bars correspond to the S.E.M. of at least two independent experiments.

The following figure supplement is available for figure 3:

**Figure supplement 1.** Resolution of RING1B/PCGF4 dimer by SDS-PAGE.

elevated H2AK119 E3 ligase activity (*Gao et al., 2012*; *Tavares et al., 2012*). In the context of our experiments, we observe a small stimulatory effect when RYBP was added to PCGF4-RING1B (*Figure 3B*). Despite this low level of stimulation, PCGF4-RING1B-RYBP was still much less active than PCGF1-RING1B-RYBP. Together these observations indicate that the PCGF component of the PRC1 complexes defines the inherent H2AK119 E3 ligase activity of individual PRC1 dimers and that inclusion of RYBP/YAF2 further matures these assemblies into robust enzymes that can efficiently modify chromatin.

## RYBP-dependent stimulation of PCGF1-PRC1 is associated with changes in the PCGF1-RING1B dimer but not with ubiquitin binding

Our in vitro reconstitution reactions demonstrated that RYBP can control how PRC1 deposits H2AK119ub1. We were therefore interested to understand the molecular nature of this stimulatory activity, as it could have profound implications for how PRC1 functions in vivo. Interestingly, RYBP has been shown to function as an ubiquitin-binding protein via its zinc finger NZF domain (*Arrigoni et al., 2006*; *Fereres et al., 2014*). We therefore reasoned that the capacity of RYBP to recognize ubiquitin, perhaps in the form of H2AK119ub1, could account for its stimulatory activity on PRC1. To examine this possibility we generated recombinant RYBP, and RYBP with mutations in the NZF domain that abrogate ubiquitin binding (*Arrigoni et al., 2006*), and added them to E3 ligase reaction containing PCGF1-RING1B (*Figure 4A and B*). Importantly, both forms of RYBP stimulated the activity of PCGF1-RING1B to a similar extent (*Figure 4B*). Given that the ability of RYBP to recognise ubiquitin did not account for its stimulation of PRC1, we wondered whether RYBP may instead affect how the core dimer assembles to function as an H2AK119 E3 ligase. We therefore turned to crosslinking mass spectrometric analyses to examine whether inclusion of RYBP in the complex resulted in any gross alterations in the proximity of individual domains. To address this possibility we directly compared the PCGF1-RING1B dimer (*Figure 4C*) and the RYBP-PCGF1-RING1B trimer (*Figure 4D*). Within the PCGF1-RING1B dimer, we observed multiple high-confidence crosslinks between the RING domain of RING1B and its C-terminal ubiquitin-like (UBL) domain,

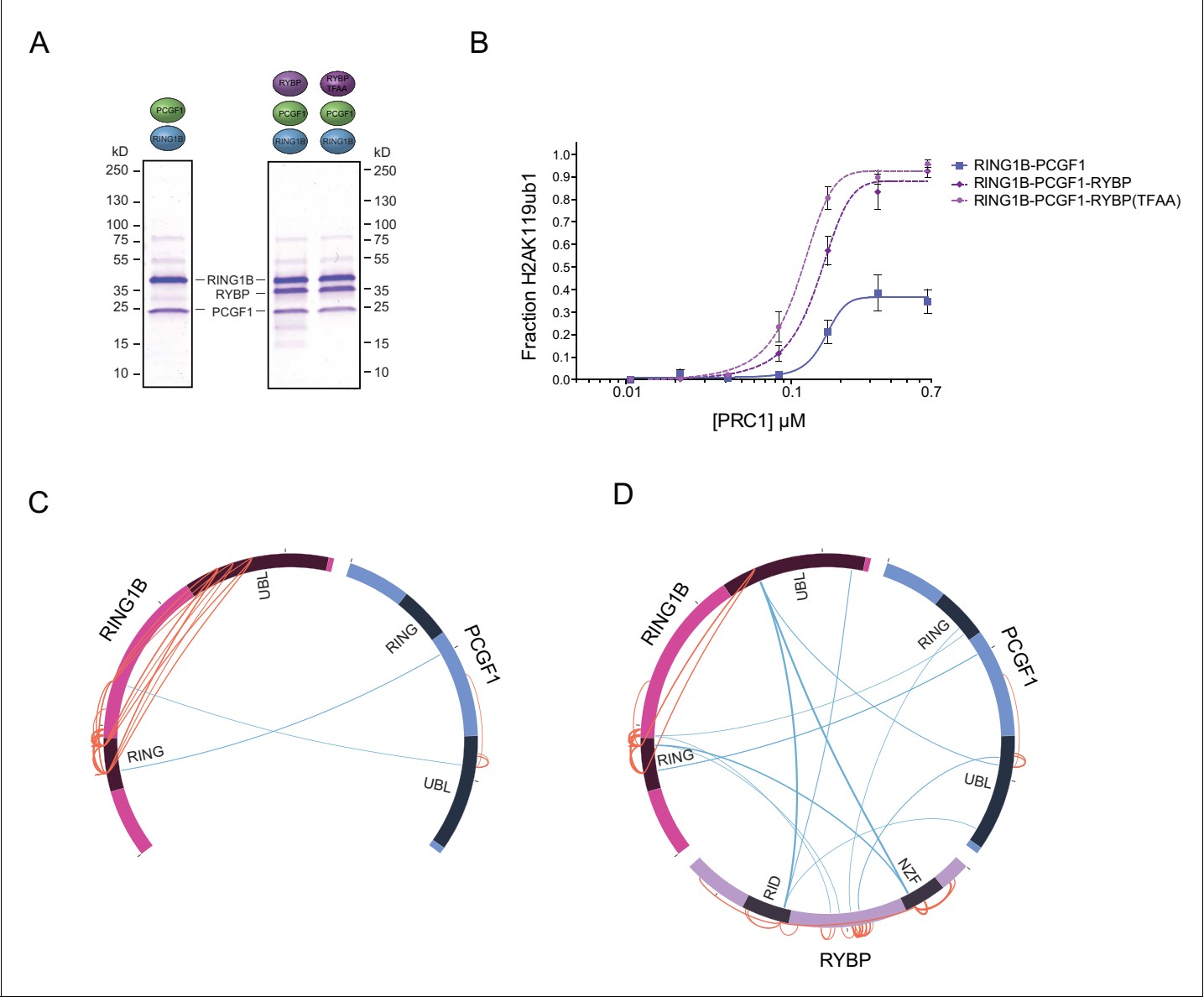

**Figure 4.** RYBP-dependent stimulation of PCGF1-PRC1 is associated with changes in the PCGF1-RING1B dimer but not ubiquitin binding. (A) Coomassie-stained gels showing the recombinant PCGF1-RING1B affinity purified via RING1B with wildtype RYBP or a ubiquitin-binding mutant (RYBP TFAA). (B) An E3 ligase activity assay comparing the activity of PCGF1-RING1B with addition of RYBP or RYBP with mutations in the NZF domain (TF-AA) that abrogate ubiquitin binding. Error bars correspond to the S.E.M. of at least two independent experiments. (C–D) Circos plots illustrating crosslinking mass spectrometry analysis of PCGF1-RING1B in presence or absence of RYBP. In (C) extensive intramolecular crosslinks are observed between the RING1B RING and UBL domains. In (D) these intramolecular crosslinks are largely ablated in the presence of RYBP which makes extensive crosslinks with these regions of RING1B. Blue and orange lines indicate intermolecular and intramolecular crosslinks respectively. Lines are weighted by the statistical confidence of their interaction.

The following figure supplement is available for figure 4:

**Figure supplement 1.** PCGF1 levels are reduced in complexes lacking RYBP.

suggesting that there are likely significant intramolecular interactions occurring between these regions (*Figure 4C*). Intriguingly, most of the RING domain/RING-UBL domain crosslinks within RING1B were lost in the trimeric complex (PCGF1-RING1B-RYBP) (*Figure 4D*), suggesting that RYBP causes an alteration in PCGF1-RING1B which limits association between its N- and C-terminal

domains, possibly to stabilize the formation of an optimally assembled and active E3 ligase. Interestingly, upon closer inspection of our reconstituted complexes (*Figure 1D*), it was noticed that the level of PCGF1 was reduced in complexes lacking RYBP (*Figure 4—figure supplement 1*). This is in agreement with our crosslinking mass spectrometry analysis, and suggests that RYBP helps to reconfigure and stabilize the PCGF1-RING1B dimer. Importantly however, reduced PCGF1 inclusion does not explain the effect that RYBP has on the activity of PCGF1/PRC1 complexes, as the presence of RYBP stimulated the relative activity of PRC1 to a much larger extent than can be simply explained by PCGF1 levels alone (*Figure 2B and C*). Furthermore, RYBP and YAF2 were able to significantly stimulate the activity of pre-existing PCGF1-RING1B assemblies (*Figure 2—figure supplement 1B*). Future structural work will be imperative to understand the defined biochemical nature of this RYBP-dependent stimulation.

## RYBP is required for H2AK119ub1 at PRC1 targets in vivo

The mechanisms that drive and shape H2AK119ub1 in vivo remain very poorly understood. Given our observation that RYBP and YAF2 play a central role in controlling PRC1 complex activity in vitro, we sought to understand whether this activity defines H2AK119ub1 in vivo. To address this possibility, we took advantage of a mouse embryonic stem cell system in which the *Rybp* gene is flanked by loxP sites and which also expresses a tamoxifen-inducible form of Cre recombinase (*Hisada et al., 2012*). However, this cell line also expresses the RYBP paralogue YAF2, which appears to stimulate PRC1 in a manner similar to RYBP (*Figure 2—figure supplement 1*), suggesting they could have redundant or overlapping functions in vivo. To circumvent this limitation, we exploited CRISPR/Cas9-based genome editing to delete two essential coding exons in *Yaf2* and create a homozygous *Yaf2$^{-/-}$* cell line that lacked *Yaf2* mRNA and protein (*Figure 5—figure supplement 1A–C* and *Figure 5A*). Treatment of the *Rybp$^{fl/fl}$;Yaf2$^{-/-}$* cell line with tamoxifen resulted in a complete loss of RYBP by 96 hr as assessed by western blot (*Figure 5B*) and at PcG targets as assessed by ChIP-seq (*Figure 5C*).

Having established the *Rybp$^{fl/fl}$;Yaf2$^{-/-}$* cell line we then set out to understand if the capacity of RYBP to stimulate PRC1 in vitro would also translate into effects on cellular H2AK119ub1 following RYBP removal. Studying this in a cellular context could however be complicated by previous conflicting observations suggesting that RING1B may, or may not, be destabilised when RYBP is removed in ESCs (*Tavares et al., 2012*; *Morey et al., 2013*). To examine this possibility we first determined whether loss of RYBP affected the level of RING1B. In the context or our new *Rybp$^{fl/fl}$;Yaf2$^{-/}$*$^{-}$conditional system, deletion of RYBP did not lead to an appreciable reduction in total RING1B (*Figure 5B*), in agreement with our observation that PRC1 complex assembly in vitro is modular (*Figure 1D*). The stability of the PRC1 complex in the absence of RYBP allowed us to then examine the direct contribution of RYBP to H2AK119ub1 in vivo. Somewhat surprisingly, despite the dramatic affects that RYBP has on H2AK119 E3 ligase activity in vitro, removal of RYBP did not significantly affect total H2AK119ub1 as measured by western blot (*Figure 5D and E*). Again this contrasts previous observations in which shRNA-mediated knockdown of RYBP led to global reductions in H2AK119ub1 (*Tavares et al., 2012*; *Gao et al., 2012*), and suggests that these affects may have been the indirect result of decreased RING1B levels.

Given these observations, we reasoned that RYBP may play a more defined or locus-specific role in regulating H2AK119ub1 as had previously been suggested from single gene analysis following RYBP depletion in ESCs (*Morey et al., 2013*). To address this possibility we exploited a calibrated ChIP-seq approach (*Bonhoure et al., 2014*; *Hu et al., 2015*; *Orlando et al., 2014*) to measure H2AK119ub1 genome-wide in the *Rybp$^{fl/fl}$;Yaf2$^{-/-}$* cells, before and after RYBP ablation. Importantly, H2AK119ub1 signal in wild-type cells was highly correlated with both RING1B and RYBP occupancy genome-wide (*Figure 5F*). Furthermore, a visual inspection of these data revealed enrichment of H2AK119ub1 at genes associated with Polycomb-mediated repression in embryonic stem cells (*Figure 5G*). When we compared the levels of H2AK119ub1 before and after RYBP removal at individual Polycomb target sites, we observed clear reductions in H2AK119ub1 (*Figure 5G*), suggesting that RYBP supports deposition of H2AK119ub1 at these genomic locations. To explore the breadth of these affects, we identified a set of 2407 genomic regions that were enriched for H2AK119ub1. Quantitation of H2AK119ub1 after removal of RYBP revealed widespread effects on H2AK119ub1 (*Figure 5H*), with approximately 45% of regions showing significant reductions in the levels of H2AK119ub1 (1091/2407; p<0.05). However, this appeared to be an underestimation of the effect,

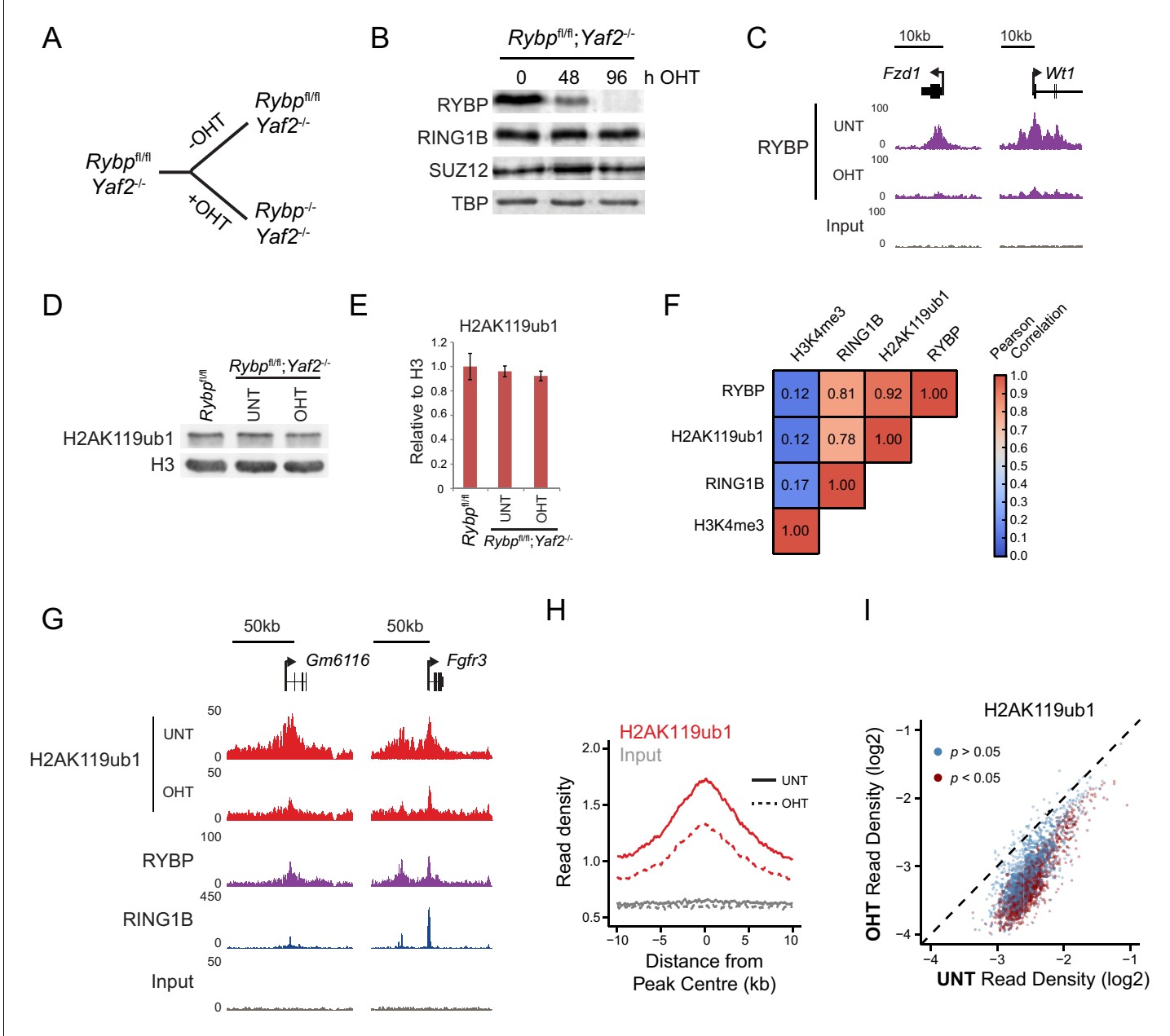

**Figure 5.** RYBP is required for H2AK119ub1 at PRC1 targets in vivo. (**A**) A schematic illustrating the *Rybp*^fl/fl^;*Yaf2*^−/−^ mouse embryonic stem cell model in which the addition of tamoxifen (OHT) leads to deletion of RYBP. (**B**) Western blot analysis demonstrating that treatment of *Rybp*^fl/fl^;*Yaf2*^−/−^ cells with tamoxifen for 96 hr results in loss of RYBP protein. The levels of the PRC1 component, RING1B, and the PRC2 component, SUZ12, are unchanged. TATA Box Binding Protein (TBP) is included as a loading control. (**C**) A genomic snapshot from RYBP ChIP-seq analysis in *Rybp*^fl/fl^;*Yaf2*^−/−^ cells before (UNT) and after 96 hr of tamoxifen treatment (OHT) showing that RYBP occupancy on chromatin is lost. (**D**) Western blot analysis of H2AK119ub1 in *Rybp*^fl/fl^ ESCs, compared with *Rybp*^fl/fl^;*Yaf2*^−/−^ cells before (UNT) and after 96 hr tamoxifen treatment (OHT). Western blot with a histone H3-specific antibody is shown as a loading control. (**E**) A quantitation of western blot analysis for H2AK119ub1 relative to histone H3 in *Rybp*^fl/fl^ ESCs, compared with RYBP/YAF2 deletion indicating no global reduction in H2AK119ub1. Errors bars indicate standard deviation for three biological replicate experiments. (**F**) Genome-wide correlation of wildtype H2AK119ub1, RYBP, RING1B and H3K4me3 ChIP-seq using 10 kb windows. H2AK119ub1 ChIP-seq correlates well with RYBP and RING1B genome-wide. (**G**) A genomic snapshot of calibrated H2AK119ub1 ChIP-seq in the *Rybp*^fl/fl^;*Yaf2*^−/−^ mouse embryonic stem cells before (UNT) and after 96 hr tamoxifen (OHT) treatment at two polycomb-occupied target sites indicating significant reductions in H2AK119ub1 in the absence of RYBP. (**H**) A metaplot illustrating calibrated H2AK119ub1 ChIP-seq read density at peaks of H2AK119ub1 (n = 2407) before (UNT-solid line) and after (OHT- dotted line) tamoxifen treatment. Removal of RYBP has a broad effect on H2AK119ub1. (**I**) A scatter plot of normalised read densities for calibrated H2AK119ub1 ChIP-seq illustrates reduced H2AK119ub1 signal at nearly all H2AK119ub1 peaks in OHT-treated *Rybp*^fl/fl^;*Yaf2*^−/−^ cells (OHT) when compared to untreated cells (UNT). Red dots correspond to statistically significant reductions in H2AK119ub1.

*Figure 5 continued on next page*

*Figure 5 continued*

The following figure supplement is available for figure 5:

**Figure supplement 1.** CRISPR-Cas9-mediateddeletion of *Yaf2* in *Rybp*<sup>fl/fl</sup> ESCs.

as nearly all PRC1 targets displayed reductions in H2AK119ub1, albeit below the significance threshold (*Figure 5I*). These effects are consistent with our in vitro observations demonstrating that RYBP shapes the E3 ligase activity of PRC1 complexes. Given the widespread reductions in H2AK119ub1 observed in ChIP-seq, it is perhaps surprising that we did not detect global alterations in H2AK119ub1 by western blot. This is likely due to a proportion of total cellular H2AK119ub1 residing outside of classical polycomb target sites as has been proposed previously in other systems (*Lee et al., 2015*; *Goldknopf et al., 1975*).

## RYBP regulates the activity, not chromatin occupancy, of PRC1 in vivo

Although our in vitro studies provide compelling evidence that RYBP regulates PRC1 activity, previous work has suggested that RYBP may also contribute to PRC1 function through regulating its stability or targeting (*Tavares et al., 2012*; *Gao et al., 2012*). Conversely, single gene studies had suggested that following depletion of RYBP, RING1B targeting to polycomb sites may not be overtly affected (*Hisada et al., 2012*; *Morey et al., 2013*). Therefore, having demonstrated that the stability of PRC1 is unaffected following RYBP deletion in the *Rybp*<sup>fl/fl</sup>;*Yaf2*<sup>−/−</sup> cells (*Figure 5B*), we set out to test at the genome-scale whether H2AK119ub1 losses following RYBP deletion in vivo were the result of defective PRC1 targeting or supressed E3 ligase activity. To achieve this, we carried out RING1B ChIP-seq before and after tamoxifen treatment of *Rybp*<sup>fl/fl</sup>;*Yaf2*<sup>−/−</sup> cells (*Figure 6*). Initially, we considered RING1B occupancy at PRC1 target sites where H2AK119ub1 was dependent upon RYBP, to examine whether H2AK119ub1 reductions were coincident with reduced chromatin occupancy of PRC1. However, visual inspection of several of these sites revealed that despite displaying large decreases in H2AK119ub1, RING1B occupancy was often unaffected (*Figure 6A*). We extended this analysis genome-wide (*Figure 6B*), and found that the majority of PRC1 target sites (78%; 1870/2407) failed to show a significant change in RING1B occupancy (*Figure 6B*) and we validated these observations at a number of sites using ChIP quantitative PCR analysis (*Figure 6—figure supplement 1*). Indeed, there were only a modest number of sites showing decreases in RING1B binding (20%; 487/2407) (*Figure 6C*). Importantly, following RYBP deletion we observed clear and widespread reductions in H2AK119ub1 at PRC1 target sites where RING1B occupancy remained unchanged. These observations are in agreement previous observations at selected target sites following shRNA-mediated knockdown of RYBP which also appeared to show loss of H2AK119ub1 in the absence of perturbed RING1B binding (*Morey et al., 2013*). Interestingly, the subset of PRC1 targets which lost RING1B appeared on average to exhibit a unique depletion of H2AK119ub1 in the centre of the peak when compared to other PRC1 targets (*Figure 6C*). This was coincident with a more highly positioned TSS at the centre of the peak but did not appear to correspond with a depletion of nucleosomes (*Figure 6—figure supplement 2A and B*). Furthermore, these sites tended to be more enriched for variant PRC1 (RYBP), as opposed to canonical PRC1 (CBX7) occupancy, suggesting that these PRC1 targets might depend more on RYBP for normal H2AK119ub1 and RING1B occupancy than canonical targets (*Figure 6—figure supplement 2C*) as suggested previously (*Morey et al., 2013*). Together, these genome-wide observations strongly support our in vitro conclusions that RYBP stimulates the E3 ligase activity of PRC1, and reveal that decreases in H2AK119ub1 at PRC1 target sites are not due to defects in targeting PRC1.

## RYBP-dependent stimulation is essential for H2AK119ub1 at sites with low PRC1 occupancy

RYBP plays a central role in stimulating PRC1 in vivo, and virtually all H2AK119ub1-enriched regions show some dependency on RYBP for normal H2AK119ub1 (*Figure 5I*). Nevertheless it was clear that at individual sites the magnitude of H2AK119ub1 loss often differed greatly, perhaps suggesting that features inherent to individual Polycomb target sites could dictate their reliance on the

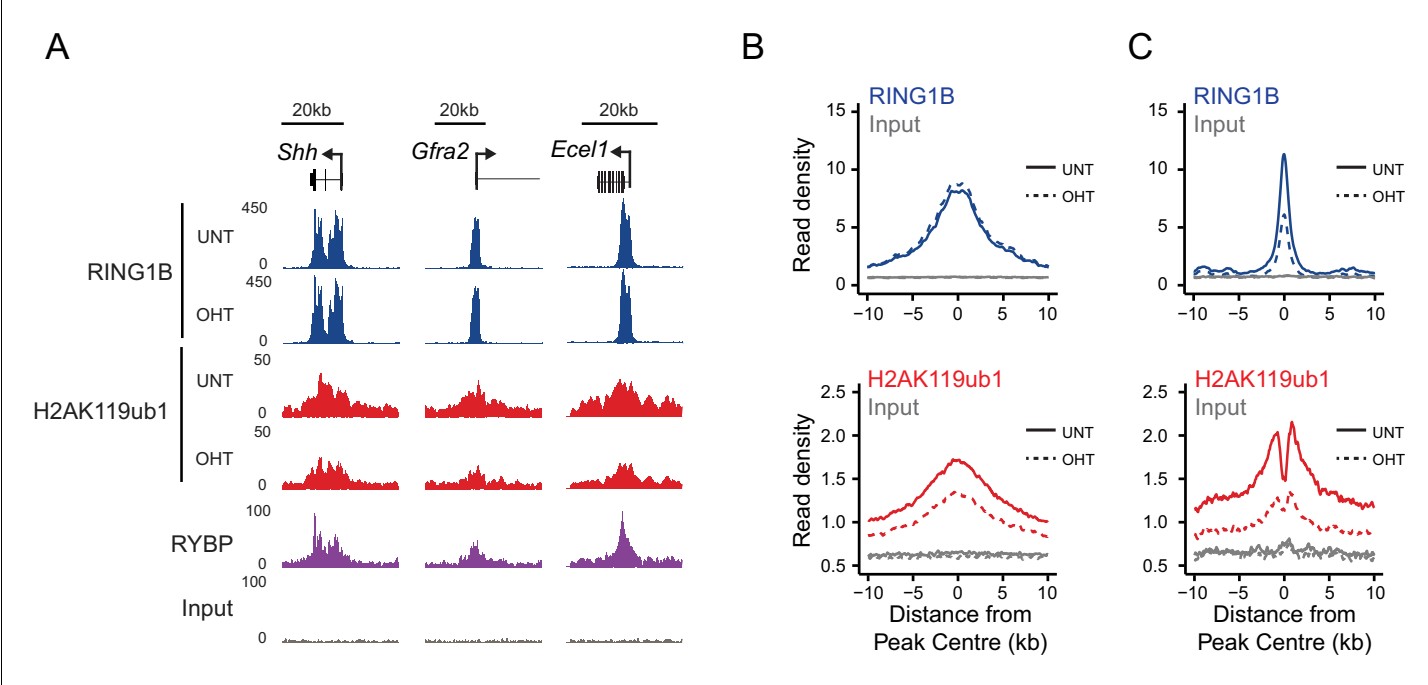

**Figure 6.** RYBP regulates the activity, not chromatin occupancy, of PRC1 in vivo. (**A**) A genomic snapshot of RING1B and H2AK119ub1 ChIP-seq in *Rybp*$^{fl/fl}$;*Yaf2*$^{-/-}$ mouse embryonic stem cells before (UNT) and after 96 hr tamoxifen (OHT) treatment demonstrates that decreases in H2AK119ub1 occur in the absence of altered RING1B occupancy. (**B–C**) Metaplots illustrating H2AK119ub1 and RING1B ChIP-seq read density at H2AK119ub1 peaks with (n = 487) or without (n = 1870) significant reductions in RING1B occupancy before (UNT-solid line) and after tamoxifen treatment (OHT-dotted line). RYBP loss affects the levels of H2AK119ub1 (B-bottom panel) even though RING1B occupancy is unaffected (B-upper panel) indicating RYBP stimulates PRC1 activity at target sites in vivo.

The following figure supplements are available for figure 6:

**Figure supplement 1.** RYBP deletion results in reductions in H2AK119ub1 at sites where PRC1 occupancy remains unchanged.

**Figure supplement 2.** An examination of genomic features at PRC1 targets that retain, or lose, RING1B following deletion of RYBP/YAF2.

stimulatory activity of RYBP. To examine this possibility we divided PRC1 target sites into quartiles based on their relative loss of H2AK119ub1 (*Figure 7A*). Initially, we examined the levels of RYBP at these different groups to determine whether the effect on H2AK119ub1 following loss of RYBP was not simply related to the absolute amount of RYBP at a given target site (*Figure 7B*). Interestingly, RYBP enrichment was similar across groups of RYBP-dependency, and counterintuitively more lowly occupied sites often exhibited more profound reductions in H2AK119ub1. We then examined the occupancy of the PRC1 catalytic core component RING1B at these same sites, and observed that the sites which were most dependent upon RYBP for H2AK119ub1 had the lowest occupancy of RING1B (*Figure 7C*). This suggests that sites with limiting amounts of PRC1 have a more defined requirement for RYBP-dependent stimulation to achieve normal H2AK119ub1. To explore this possibility, we calculated the ratio of RYBP relative to RING1B at PRC1 target sites over the range of H2AK119ub1 dependencies (*Figure 7D*). This revealed that the sites where loss of RYBP most profoundly affected H2AK119ub1 exhibited the highest ratios of RYBP to RING1B (*Figure 7D and E*). At these PRC1 target sites it appears that the inherent activity of the core catalytic complex is insufficient to sustain H2AK119ub1, and that RYBP-dependent stimulation is required to define normal H2AK119ub1 levels. We then wondered whether dependency on RYBP might also be related to the occupancy of canonical PRC1 at these sites. Indeed, PRC1 targets that were occupied by RYBP but displayed reduced levels of CBX7 (i.e. 'variant-enriched') tended to show the largest reductions in H2AK119ub1 following RYBP removal (*Figure 7F and G*), whereas variant and canonical PRC1 occupied sites (i.e. 'shared') appeared to rely on RYBP to a lesser extent. Intriguingly, shared-PRC1

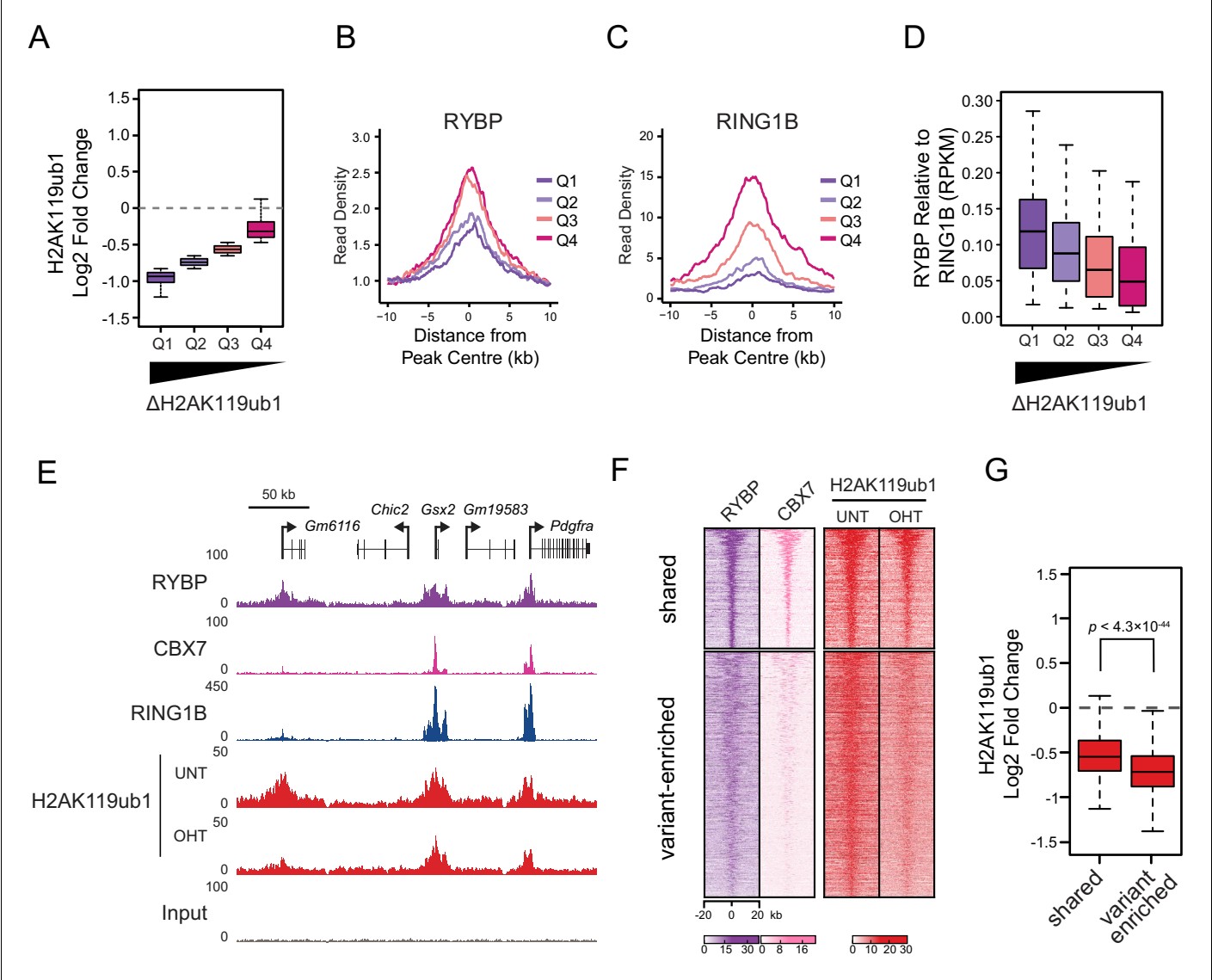

**Figure 7.** RYBP-dependent stimulation is essential for H2AK119ub1 at sites with low PRC1 occupancy. (**A**) A box plot illustrating the separation of H2AK119u1 peaks into quartiles based on their relative reduction in H2AK119ub1 upon deletion of RYBP. This indicates that all H2AK119ub1 peaks experience a decrease in H2AK119ub1 following RYBP removal, but some sites (e.g. Q1) are more dependent upon RYBP than others (e.g. Q4) (**B–C**) Metaplots illustrating the RYBP and RING1B ChIP-seq read density over the H2AK119ub1 quartiles identified in (**A**). Quartiles that are most dependent on RYBP for H2AK119ub1 (i.e. Q1 and Q2) have lower RYBP and RING1B occupancy. (**D**) A box plot illustrating that the quartiles which are most dependent on RYBP have the largest enrichment of RYBP relative to RING1B. (**E**) A genomic snapshot depicting three polycomb target sites which display varying dependencies on RYBP for their H2AK119ub1. The *Gm6116* gene (highlighted in grey) has low occupancy of RING1B and CBX7, and is almost completely reliant on RYBP for H2AK119ub1. In contrast the *Gsx2* and *Pdgfra* genes have high RING1B and CBX7 occupancy and are less dependent on RYBP for their H2AK119ub1. This indicates that RYBP plays a more pronounced role stimulating PRC1 activity at sites with limiting levels of RING1B. Furthermore, sites that are highly dependent on RYBP appear to be depleted of canonical PRC1. (**F**) A heatmap illustrating H2AK119ub1 peaks clustered into variant-enriched PRC1 and variant and canonical PRC1 (shared) sites based on RYBP (variant PRC1) and CBX7 (canonical PRC1) levels. Both variant-enriched and shared PRC1 target sites show a reduction in H2AK119ub1 following deletion of RYBP. (**G**) A box plot indicating the alterations in H2AK119ub1 at shared or variant-enriched PRC1 target sites. The *p* value denotes statistical significance calculated by a Wilcoxon signed rank test.

The following figure supplement is available for figure 7:

**Figure supplement 1.** Gene ontology analysis of shared and variant-enriched PRC1 target sites.

targets, tended have the largest polycomb domains, and were enriched for genes related to sequence-specific transcription factor binding (*Figure 7—figure supplement 1*), perhaps reflecting a contribution of both variant and canonical PRC1 in shaping H2AK119ub1 at these regulatory genes. Interestingly, and in contrast to previous reports (*Morey et al., 2013*), our ChIP-seq based analysis of RYBP binding did not reveal a class of PcG target sites that were enriched for canonical PRC1 (CBX7) but lacked variant PRC1 (RYBP). This suggests that canonical PRC1 binding occurs at PcG target sites that are also occupied by variant PRC1. Together, our detailed genome-wide analysis in vivo establishes a clear role for RYBP in stimulating the H2AK119ub1 E3 ligase activity of PRC1, particularly at variant PRC1-enriched sites with low levels of RING1B.

## RYBP-dependent H2AK119ub1 shapes PRC2 activity

We and others have recently discovered that PRC1 can regulate the targeting of PRC2 through its capacity to recognize H2AK119ub1 (*Blackledge et al., 2014*; *Cooper et al., 2014*; *Kalb et al., 2014*). This, coupled with CBX-dependent recognition of H3K27me3 by canonical PRC1 complexes (*Bernstein et al., 2006*; *Fischle et al., 2003*; *Min et al., 2003*), could effectively create a feedback mechanism sufficient to help sustain Polycomb chromatin domains in a histone modification-dependent manner. Given that RYBP stimulates the E3 ligase activity of PRC1, we wondered whether this activity may play a role in these proposed feedback mechanisms. To address this we performed ChIP-seq for SUZ12, a core component of PRC2, and H3K27me3 in the $Rybp^{fl/fl};Yaf2^{-/-}$ cells before and after tamoxifen treatment (*Figure 8A*). Strikingly, deletion of RYBP resulted in genome-wide reductions in H3K27me3 at Polycomb target sites (*Figure 8A and B*). However, like H2AK119ub1, H3K27me3 reductions were not evident from western blot analysis of total histones, suggesting that H3K27me3 is affected most profoundly at classical Polycomb target sites and not the larger pool of H3K27me3 which is proposed to exist elsewhere in the genome (*Figure 8—figure supplement 1*) (*Grzybowski et al., 2015*). Importantly, H3K27me3 reductions at Polycomb target sites occurred without any apparent loss in the chromatin occupancy of the PRC2 component SUZ12 (*Figure 8A and C*, *Figure 8—figure supplement 2*), suggesting that the observed decreases in H3K27me3 likely arise as a result of reduced activity of the PRC2 complex. Like the observed changes in H2AK119ub1 (*Figure 7F and G*), the largest reductions in H3K27me3 were evident at variant-enriched PRC1 target sites (*Figure 8D*), consistent with a more defined role for RYBP in maintaining PcG-associated histone marks at these regions. Importantly, reductions in H3K27me3 across all PRC1 target sites correlated closely with reductions in H2AK119ub1 (*Figure 8E*), indicating that the effects on H2AK119ub1 and H3K27me3 are linked and directly related to the activity of RYBP. These important observations demonstrate that RYBP-dependent H2AK119ub1 shapes the activity of PRC2 in depositing H3K27me3, and suggests that the observed stimulation of PRC2 by nucleosomes containing H2A mono-ubiquitylation in vitro is also relevant in vivo (*Kalb et al., 2014*).

Given that PRC1 and PRC2 complexes recognize and bind to the histone modifications that they place, it was somewhat surprising to us that the reductions in H2AK119ub1 and H3K27me3 following RYBP/YAF2 deletion did not result in more widespread effects on PRC1 and PRC2 occupancy. One possible explanation is that normal PcG protein occupancy can only be effectively maintained as long as histone modifications remain above a certain level or threshold. Conversely, if this threshold is crossed, effective Polycomb chromatin occupancy may no longer be maintained, and the Polycomb chromatin domain could begin to erode. If this were the case, one might predict that thresholds for Polycomb domain erosion would differ between individual target sites. We therefore hypothesized that in the context of our experiments, erosion of Polycomb chromatin domains following RYBP deletion might occur at the sites that experienced the greatest relative reductions in their Polycomb-associated histone modifications. To examine whether this was the case, we examined the subset of Polycomb target sites that experienced significant losses in PRC1 and PRC2 occupancy following RYBP deletion, and compared these to target sites at which occupancy was unaffected. Interestingly, we observed that the sites with the most profound losses of H2AK119ub1 and H3K27me3 were the same sites that showed reductions in PRC1 and PRC2 (*Figure 8F*). Together these observations reveal two important and fundamental features associated with the function of Polycomb chromatin domains. Firstly, there is an intimate relationship between the histone-modifying activities of PRC1 and PRC2 at target sites, which is regulated by auxiliary PcG complex subunits and not simply dictated by occupancy of PRC1/PRC2 core catalytic subunits. This is evident in instances where RYBP removal leads to reductions in both H2AK119ub1 and H3K27me3, without major alterations in

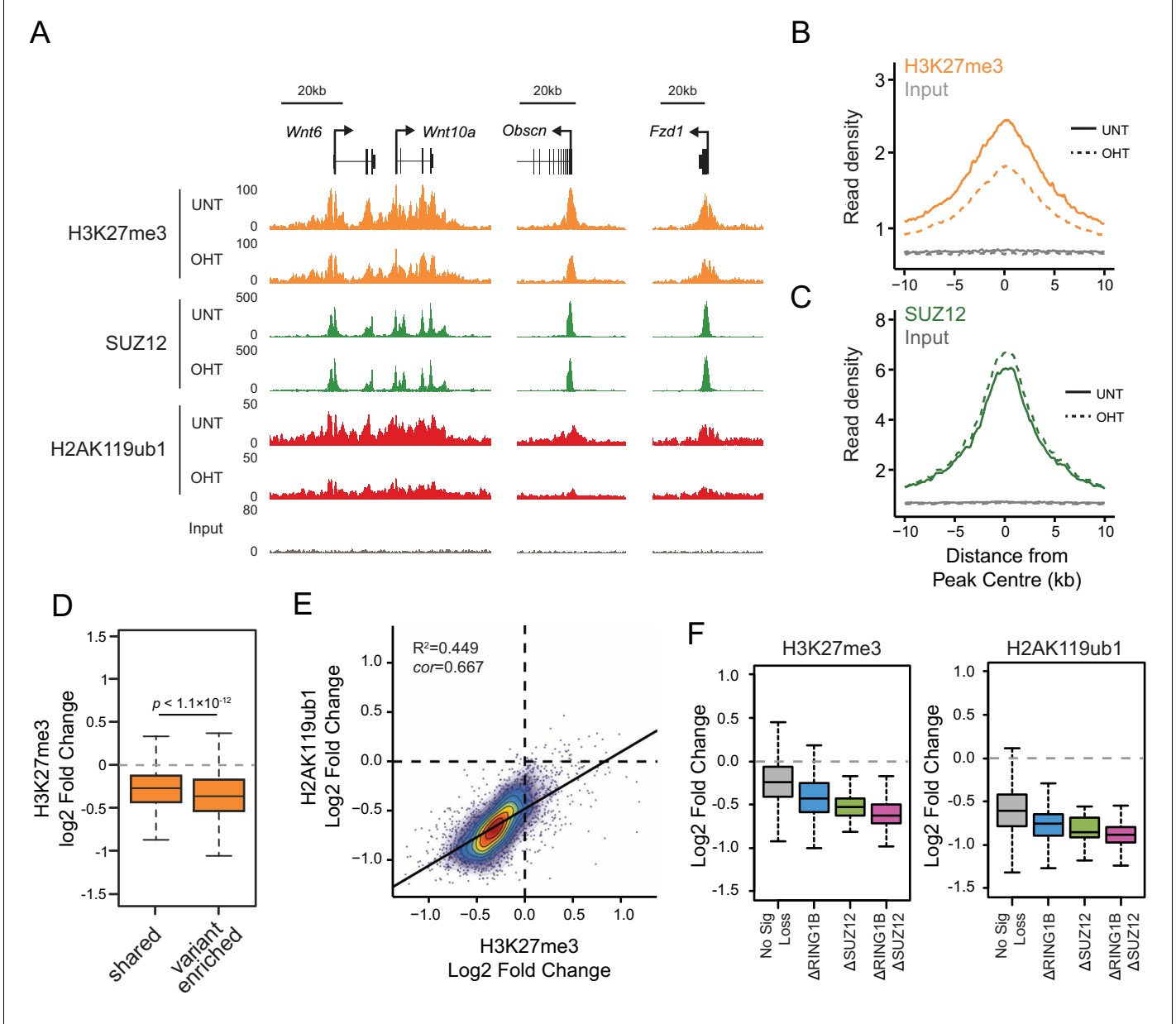

**Figure 8.** RYBP-dependent H2AK119ub1 shapes PRC2 activity. (**A**) A genomic snapshot of H3K27me3, SUZ12 and H2AK119ub1 ChIP-seq in *Rybp^fl/fl*; *Yaf2^−/−* mouse embryonic stem cells before (UNT) and after tamoxifen (OHT) treatment. At these target sites there are reductions in H3K27me3 in the absence of changes in SUZ12 occupancy following RYBP deletion. (**B-C**) Metaplots illustrating ChIP-seq read density for H3K27me3 (**B**) and SUZ12 (**C**) in the *Rybp^fl/fl*;*Yaf2^−/−* mouse embryonic stem cells before (UNT-solid line) and after tamoxifen (OHT-dotted line) treatment. Removal of RYBP results in widespread loss of H3K27me3 despite largely unaffected occupancy of SUZ12. (**D**) A box plot indicating the alterations in H3K27me3 at shared or variant-enriched PRC1 target sites. The *p* value denotes statistical significance calculated by a Wilcoxon signed rank test. (**E**) A scatter plot showing changes in H2AK119ub1 and H3K27me3 normalised read densities at H2AK119ub1 peaks after deletion of RYBP. This illustrates that the fold change in H2AK119ub1 and H3K27me3 are related to each other at individual sites following loss of RYBP. $R^2$ and c*or* denote coefficient of determination for linear regression model and the pearson correlation coefficient respectively. (**F**) A box plot illustrating the changes in H2AK119ub1 and H3K27me3 at H2AK119ub1 peaks divided into groups based on whether RYBP deletion resulted in no significant loss of RING1B or SUZ12 (No Sig Loss, n = 1947), loss of RING1B only (△RING1B, n = 319), loss of SUZ12 only (△SUZ12, n = 38) or loss of both RING1B and SUZ12 (△RING1B △SUZ12, n = 103). This indicates that sites susceptible to loss of RING1B and/or SUZ12 experience the greatest relative reductions in H2AK119ub1 and H3K37me3.

The following figure supplements are available for figure 8:

**Figure supplement 1.** Analysis of H3K27me3 levels following loss of RYBP/YAF2.

*Figure 8 continued on next page*

*Figure 8 continued*

**Figure supplement 2.** RYBP deletion results in reductions in H3K27me3, without decreases in PRC2 occupancy.

PcG complex occupancy. Secondly, the occupancy of PcG complexes at target sites is not driven in a linear fashion by H2AK119ub1 and H3K27me3 levels. However, if activity-based communication between the PcG complexes is compromised there appears to be a modification threshold below which complex occupancy can no longer be sustained. Again this is evident following RYBP removal, where sites displaying the greatest loss of H2AK119ub1 and H3K27me3 also show reductions in PcG complex occupancy.

## Loss of RYBP culminates in gene reactivation at sites where Polycomb chromatin domains are compromised

Studies in mice have demonstrated that RYBP plays a fundamental role in early development (*Pirity et al., 2005*), and that it is also important at later developmental stages including haematopoiesis (*Calés et al., 2016*), presumably due to its involvement in PcG-mediated gene regulation (*Tavares et al., 2012*). Furthermore, RYBP-deficient *Drosophila* die at varying stages during development and RYBP can function as an enhancer of PcG phenotypes (*Gonzalez et al., 2008*). Given the widespread effects that RYBP has on shaping H2AK119ub1 and H3K27me3, and the erosion of normal Polycomb chromatin domains that ensues at some target sites following RYBP removal, we wondered whether genes associated with these target sites were aberrantly transcribed. To this end, we isolated nuclear RNA from cells before and after RYBP deletion and carried out RNA-seq analysis. This revealed that loss of RYBP resulted in a limited number of robust and statistically significant transcriptional changes (*Figure 9A*), in agreement with previous studies examining gene expression by microarray analysis in RYBP deleted cells (*Hisada et al., 2012*). Consistent with a role for RYBP in shaping PRC1-dependent gene repression, more genes were significantly upregulated (n = 230) than downregulated (n = 37), and these tended to be lowly expressed genes (*Figure 9A*). Furthermore, when we compared these gene expression changes to those observed in the complete absence of PRC1 activity (RING1A$^{-/-}$;RING1B$^{-/-}$) (*Schoenfelder et al., 2015*), most RYBP-repressed genes fell within the set of genes that are also regulated by RING1A/B (*Figure 9B*). In addition, when we compared the expression of genes that were upregulated following RYBP loss to the gene expression changes in cells where RING1B was catalytically inactivated (RING1B I53A) (*Illingworth et al., 2015*), RYBP-dependent genes were also partially upregulated, although the effects in this context were modest (*Figure 9—figure supplement 1*). The extent of gene reactivation at Polycomb target genes following RYBP removalwas generally of lower magnitude than gene reactivation following deletion of RING1A/RING1B (*Figure 9C*), in agreement with our observations that RYBP stimulates, but is not absolutely required for, PRC1 activity and occupancy. Furthermore, both shared and variant-enriched PRC1 target sites were subject to reactivation following removal of RYBP (*Figure 9—figure supplement 2*).

Having identified the genes that were most dependent on RYBP for transcriptional repression, we were keen to examine how these effects on transcription were related to any associated Polycomb chromatin domains. We reasoned that if the integrity of a Polycomb chromatin domain was required for normal transcriptional repression, one might expect that gene reactivation following RYBP loss would correspond to sites where there was a loss of PcG complex occupancy. Indeed, when we examined individual RYBP-dependent genes, increased transcription was often coincident with TSS-associated decreases in PcG complex occupancy (*Figure 9D*), although this was more apparent for PRC1 than PRC2 (*Figure 9E*; *Figure 9—figure supplement 3*). When we extended this analysis genome-wide there was a very clear relationship, albeit not perfectly linear, between reduction of PRC1 occupancy at TSSs and increased gene expression (*Figure 9F*). Consistent with a role for PcG systems in helping to maintain genes in a repressed state in tissues where they should not normally be expressed (*Klose et al., 2013*; *Riising et al., 2014*; *Voigt et al., 2013*), the magnitude of gene reactivation events was often modest. This likely reflects genes that are in a very lowly transcribed or off state, transitioning to a more transcribed but not fully activated state. This is presumably because

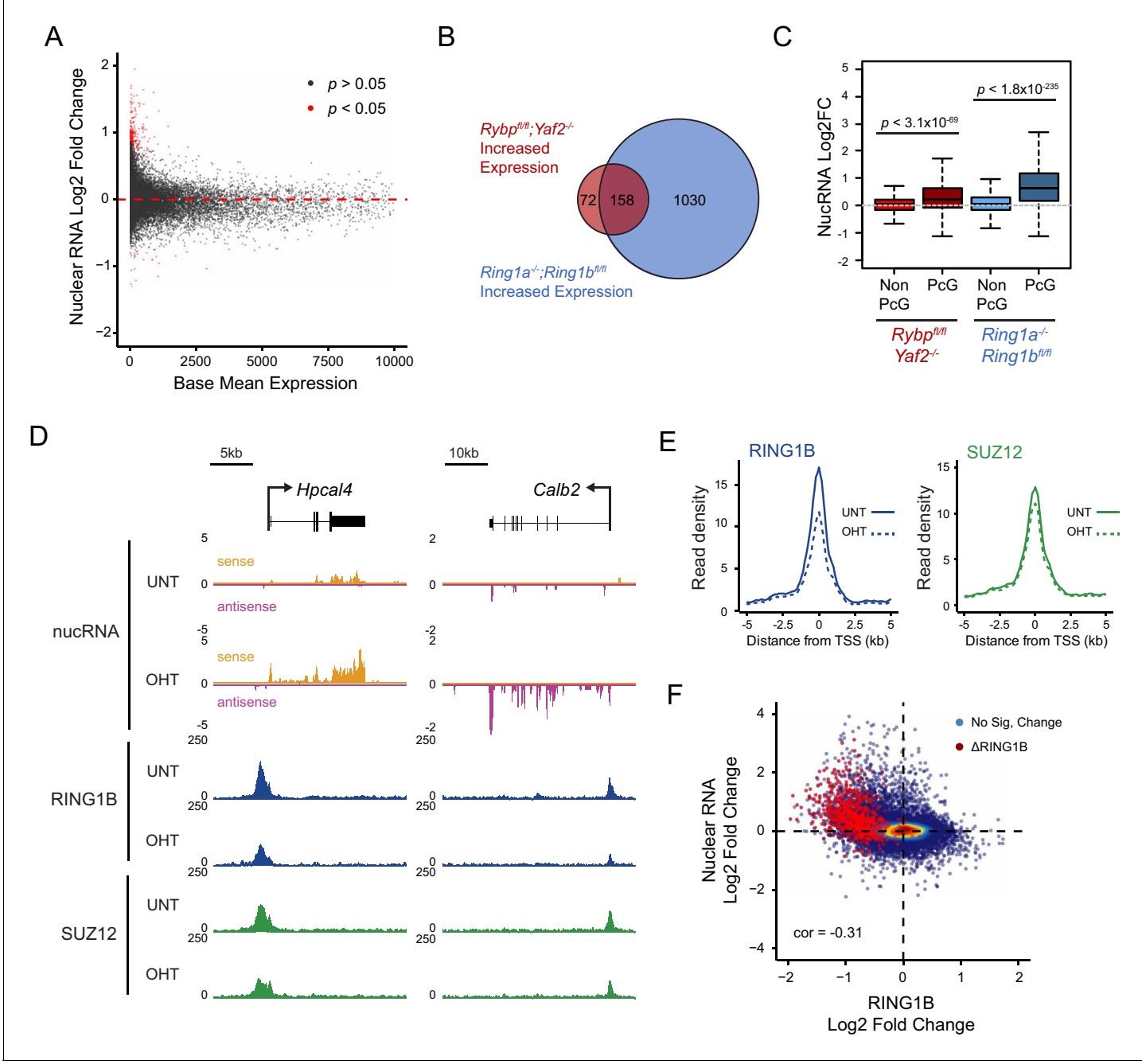

**Figure 9.** Loss of RYBP culminates in gene reactivation at sites where Polycomb chromatin domains are compromised. (**A**) An MA plot illustrating differential gene expression analysis based on nuclear RNA (nucRNA)-seq in the *Rybp*<sup>fl/fl</sup>;*Yaf2*<sup>−/−</sup> mouse embryonic stem cells before and after tamoxifen treatment. Log2 fold change of nucRNA is plotted against mean transcriptional level in wildtype cells. Significance values represent adjusted *p* values calculated by DESeq2. (**B**) A Venn diagram illustrating that genes upregulated after deletion of RYBP are a subset of genes upregulated upon deletion of the PRC1 catalytic core proteins RING1A/RING1B. (**C**) A box plot comparing the log2 fold change in nucRNA levels following loss of RYBP or RING1A/RING1B at TSS marked by H2AK119ub1 peaks (PcG; n = 1490) or TSS without H2AK119ub1 (Non-PcG; n = 12,911). This reveals that Polycomb target TSS are the most affected genes, and that the average level of upregulation after loss of RYBP is less than the upregulation following deletion of RING1A/RING1B. *p* values denote statistical significance calculated with Wilcoxon signed rank test. (**D**) A genomic snapshot of two genes upregulated following loss of RYBP. RNA-seq reads are separated by strand. Following RYBP deletion, both genes exhibit reductions in TSS-associated RING1B and SUZ12, as illustrated by ChIP-seq for RING1B and SUZ12 before (UNT) and after tamoxifen (OHT) treatment. (**E**) Metaplots illustrating that TSS occupancy of RING1B and SUZ12 are reduced after deletion of RYBP/YAF2 at upregulated genes (n = 230). (**F**) A scatter plot illustrating that increases in gene expression broadly correlate with decreases in RING1B occupancy at the TSS (±500 bp). Significant reductions in RING1B are depicted as red dots.

*Figure 9 continued on next page*

*Figure 9 continued*

The following figure supplements are available for figure 9:

**Figure supplement 1.** A Comparison of RYBP-dependent gene expression changes with RING1B[I53A] gene expression changes.

**Figure supplement 2.** Genes up-regulated following deletion of RYBP/YAF2 are not unique to shared or variant-enriched PRC1 target sites.

**Figure supplement 3.** Genes that are reactivated in response to RYBP deletion exhibit polycomb domain erosion.

the signals necessary to drive transcriptional activation of these genes are low or absent in embryonic stem cells.

Together, our transcription and Polycomb chromatin domain analyses reveal an activity-based communication between PcG complexes regulated by RYBP-dependent stimulation of PRC1. Furthermore it provides clear evidence that the histone modifying activities of PRC1 and PRC2 underpin feedback mechanisms that are important for maintaining Polycomb chromatin domains (*Blackledge et al., 2015*), counteracting inappropriate gene activation signals, and shaping normal gene repression, at least at a subset of polycomb target sites.

## Discussion

PcG proteins play an important chromatin-based role in controlling gene expression. The PRC1 and PRC2 complexes work together to achieve this, and in vertebrates both their occupancy on chromatin and functionality appear to rely, at least in part, on their capacity to modify histones. However, mechanisms controlling the activity of PRC1, and its relationship with PRC2, remain poorly understood. Here we reveal a defined requirement for specific PCGF components and the auxiliary subunits RYBP/YAF2 in regulating the H2AK119 E3 ligase activity of PRC1 in vitro (*Figures 1–4*). In vivo, RYBP plays a central role in shaping normal H2AK119ub1 at Polycomb target sites (*Figure 5*) where it functions to stimulate the E3 ligase activity of PRC1 (*Figures 6 and 7*) and regulate PRC2-dependent H3K27me3 (*Figure 8*). At a subset of Polycomb target sites, the absence of RYBP leads to a more profound effect on histone modifications leading to Polycomb chromatin domain erosion (*Figure 8*) and inappropriate reactivation of gene expression (*Figure 9*). Together this extends our understanding of the interesting and important activity-based communication between PcG complexes that has been previously proposed from in vitro studies (*Kalb et al., 2014*), and demonstrates that the enzymatic activity of PcG complexes can communicate through chromatin in vivo in order to maintain normal PcG function in gene regulation.

A striking and central conclusion from our enzymatic analysis of PRC1 complexes in vitro was that the enzymatic activity of PRC1 is determined not only by its interaction with auxiliary factors like RYBP/YAF2 but also the PCGF protein that forms the PCGF/RING1 catalytic core. This therefore suggests that the activity of individual PRC1 complexes is likely to depend upon the inclusion of specific PCGF proteins and to be highly regulated in vivo. An important focus for future work will be to directly compare the E3 ligase activities of other less well-characterized PRC1 complexes, to understand how widespread these regulatory mechanisms are amongst distinct PRC1 assemblies, including those PRC1 complexes formed by PCGF3/5/6. In an important first step towards achieving this, a recent study examined PCGF1–6/RING1 catalytic dimers and concluded that they displayed little inherent difference in their H2AK119 E3 ligase activity on nucleosome substrates in vitro (*Taherbhoy et al., 2015*). However, these experiments relied on truncated proteins which lack important domains, including the C-terminus of RING1 which is the interaction site for RYBP/YAF2 (*Wang et al., 2010*). In contrast our observations using full length proteins demonstrate that PRC1 activity is governed by the nature of the catalytic dimer and the stimulatory activity of auxiliary factors like RYBP/YAF2. An important focus of future work will also rely on understanding the relative contribution of individual PRC1 complexes in placing H2AK119ub1 in vivo, given that RYBP/YAF2 appears to interact with diverse PRC1 complexes.

As we begin to learn more about the mechanisms that shape PRC1-dependent E3 ligase activity, our understanding of how this contributes towards PcG system function in gene regulation and

development remains an important and outstanding question. Indeed, it still remains incompletely defined how the H2AK119ub1-dependent and independent activities of PRC1 are leveraged by the PcG systems to regulate gene expression, particularly in mammals. However, early biochemical work very clearly demonstrated a structural role for PRC1 in chromatin compaction and counteracting chromatin remodelling in vitro, independently of H2AK119ub1 (*Grau et al., 2011*; *Francis et al., 2004*). Furthermore, RING1B null cells which display transcriptional reactivation and chromatin decompaction of the PcG-occupied *Hox* loci, are rescued of these deficiencies with a form or RING1B lacking catalytic activity, suggesting a structural role on chromatin for PRC1 in this context (*Eskeland et al., 2010*). In agreement with a role for canonical PRC1 in shaping chromatin structure at PcG target sites, the PHC component of canonical PRC1 can polymerize in vitro and this activity appears to be import for the formation of cytologically identifiable structures referred to as 'polycomb bodies' (*Kim et al., 2002*; *Isono et al., 2013*). Similar studies in *Drosophila* have also provided evidence via super resolution imaging that Polycomb body formation relies on Ph (orthologous to mammalian PHC) to shape chromatin domains (*Boettiger et al., 2016*; *Wani et al., 2016*) and perturbation of Ph in *Drosophila* leads to widespread effects on normal gene repression (*Lagarou et al., 2008*). This abundance of work clearly indicates that PRC1 has essential structure-based effects on chromatin that are related to normal PcG-dependent gene regulation and in some cases this appears to be independent of the E3 ligase activity of PRC1. Consistent with H2AK119ub1-independent roles for PRC1 in normal *Drosophila* development, a catalytic mutant of the *Drosophila* RING1 orthologue (Sce1) is less phenotypically severe than that of Sce1 null animals (*Pengelly et al., 2015*). Furthermore, mutation of histone H2A such that it is no longer a target for PRC1-dependent mono-ubiquitylation did not lead to *Hox* gene misregulation in *Drosophila* imaginal disc clones, suggesting that H2A ubiquitylation is not required to maintain repression of these PcG target genes (*Pengelly et al., 2015*). Similarly a *Ring1b^{I53A/I53A}* mouse model, in which the E3 ligase activity of RING1B-containing PRC1 complexes is perturbed, indicated that the catalytic activity of RING1B is also not essential for early mouse development, with *Ring1b^{I53A/I53A}* mice succumbing to embryonic lethality at a later embryonic stage than *Ring1b*-null mice (*Illingworth et al., 2015*; *Voncken et al., 2003*). Because these *Ring1b* mouse models do not completely phenocopy each other it would suggest that RING1B may contribute functions during mammalian development that are independent of H2A mono-ubiquitylation, possibly related to its role in chromatin compaction (*Eskeland et al., 2010*; *Francis et al., 2004*). However, it still remains unclear whether early mammalian development and normal polycomb mediated repression can be sustained in the absence of PRC1 E3 ligase activity, as this requires the generation of a mouse model that lacks the enzymatic activity of RING1A and RING1B, both of which are expressed in early development and contribute to deposition of H2AK119ub1 and gene repression (*de Napoles et al., 2004*; *Endoh et al., 2008*). Interestingly, however, *Ring1b^{I53A/I53A}* embryonic stem cells have dramatically reduced levels of H2AK119ub1, show reductions in H3K27me3 and reactivate a subset of PcG target genes (*Illingworth et al., 2015*). This scenario is reminiscent of the reductions in H2AK119ub1 and H3K27me3 that occur following loss of RYBP-dependent stimulation of PRC1. Together these observations support the general argument that in mammals H2AK119ub1 helps to shape communication between PRC1 and PRC2 in order to sustain normal Polycomb chromatin domain function in gene regulation. However it is clear that a more detailed analysis of both RING1A and RING1B activity will be imperative in defining how important H2AK119ub1 is for mammalian development and gene regulation by the PcG system.

PRC1 and PRC2 complexes have the capacity to recognise H2AK119ub1 and H3K27me3, and it has been proposed that this supports PcG protein binding to chromatin (*Blackledge et al., 2015*; *Kalb et al., 2014*; *Margueron et al., 2009*; *Min et al., 2003*; *Wang et al., 2004b*; *Simon and Kingston, 2013*). Here we show that RYBP deletion causes reductions in H2AK119ub1 and H3K27me3 yet PRC1 and PRC2 occupancy remains largely intact. However, a subset of PcG target sites exhibit larger reductions in H2AK119ub1 and H3K27me3 and have pronounced losses of PcG complex occupancy. Based on these observations we speculate that activity-based communication between PRC1 and PRC2 may support PcG complex occupancy through a histone modification-dependent thresholding mechanism. In the context of such a mechanism we speculate that the majority of PcG target genes have H2AK119ub1 and H3K27me3 levels that are in excess of that required to support normal PRC1 and PRC2 binding. However, if H2AK119ub1 and H3K27me3 levels fall below a given threshold, as some sites do in the absence of RYBP, normal activity-based communication between

PRC1 and PRC2 can no longer be maintained, leading to a reduction in PcG protein occupancy and an increased propensity for gene reactivation.

If histone modification-dependent thresholding mechanisms contribute at least at some sites to the function of the Polycomb system, how or why might this be beneficial to the control of gene expression? Given the highly dynamic nature of PRC1 and PRC2 interactions with chromatin (*Fonseca et al., 2012*), histone modifications could provide a form of local epigenetic memory to bridge the intervening time between individual engagement events to reinforce PcG protein occupancy. Assuming that PcG protein occupancy is important for gene repression and that histone modifications can stabilize PcG protein binding, it would seem sensible that some fluctuation in histone modification levels would need to be tolerated in order to protect the stability of the PcG system at individual sites. However, this hysteresis could be rapidly counteracted by altering the activity of individual PcG complexes, or by removing their associated histone modifications. In both scenarios this would impair PcG complex occupancy and render these normally inactive states reversible and responsive to strong gene activation signals during cell lineage commitment. In combination with chromatin-modifying activities of the Trithorax group systems that are associated with gene activation, this could support a form of chromatin bistability at gene regulatory elements, as we and others have previously proposed, to allow switch-like activation of gene expression (*Deaton and Bird, 2011*; *Klose et al., 2013*; *Steffen and Ringrose, 2014*; *Voigt et al., 2013*). Interestingly, alterations in the histone modifying activities of PcG complexes are key molecular defects associated with various cancers, suggesting that perturbing PcG-associated histone modifications, or potentially altering histone modification thresholds, may be important in the pathological adaptation of gene expression in cancer (*Scelfo et al., 2015*). Clearly more work is required to understand the relative contributions of the histone modification-dependent and -independent activities of PcG complexes to both chromatin structure and gene repression at polycomb chromatin domains.

In conclusion, our enzymatic analysis of PRC1 in vitro and genomic dissection of how RYBP shapes its activity in vivo, reveal an interesting activity-based communication between Polycomb repressive complexes on chromatin. Furthermore, they indicate that histone modification thresholds could play an important and previously unrealized role in shaping Polycomb chromatin domain function in the context of gene regulation.

## Materials and methods

### Protein expression and purification

Mouse *Ring1b*, *Pcgf1*, *Rybp*, *Bcor* and *Kdm2b* (short isoform) were cloned into the Multibac plasmid set. Acceptor and donor plasmids were recombined using Cre recombinase (NEB, Ipswich, MA) as previously described (*Bieniossek et al., 2012*) and transfected into SF21 cells to make infectious baculovirus particles. Baculoviruses were then used to express proteins in SF21 cells. Baculovirus expressed proteins were then extracted and affinity purified via a FLAG-tagged RING1B subunit on FLAG affinity resin (Sigma, St Louis, MO) and eluted under native conditions using a 1x FLAG peptide (GL Biochem, Shanghai, China) with the lysis buffer containing 10 mM Tris (pH 8.0), 500 mM NaCl, 4 mM $MgCl_2$, 2 mM DTT, 20% glycerol (v/v), cOmplete Protease Inhibitor Cocktail (Roche, UK), wash buffer containing 20 mM HEPES (pH 7.9), 150 mM NaCl, 2 mM $MgCl_2$, 1 mM DTT, 15% glycerol (v/v), cOmplete Protease Inhibitor Cocktail, dilution buffer containing 10 mM Tris (pH 8.0), 10% glycerol (v/v), 0.02 % NP-40, and elution buffer consisting of wash buffer with 0.2 mg/mL 1x FLAG peptide. Mouse *Rybp* and *Yaf2* were cloned into pNIC28 and transformed into *E. coli* Rosetta2 cells; mutagenesis of *Rybp* was carried out using the QuikChange II mutagenesis kit (Agilent). His-tagged proteins were purified on Co-Talon resin using the same buffers as for FLAG purification, but with the exception of DTT and the addition of 10 mM imidazole to the wash buffer, and 150 mM imidazole to the elution buffer instead of FLAG peptide. This was followed by anion exchange chromatography (HiTrap Q FF; GE Healthcare, UK).

### Crosslinking mass spectrometry

Protein crosslinking was carried out as previously described (*Leitner et al., 2014*). 30 μg of PRC1 protein complex was crosslinked with BS3 (Creative Molecules) for 30 min at 37°C. The concentration of BS3 was optimised for each complex to prevent over-crosslinking as analysed by SDS-PAGE.

Crosslinking reactions were quenched with 1 M $NH_4HCO_3$ to a final concentration of 50 mM and incubated at 37°C for 20 min. Crosslinking reactions were then evaporated to near-dryness in a vacuum centrifuge, and urea was then added to a final concentration of 6 M. Disulfide bonds were reduced with TCEP for 30 min at 37°C, followed by iodoacetamide treatment for 30 min at room temperature in the dark, and the reaction mixture was then diluted with 50 mM $NH_4HCO_3$ to a final concentration of 1 M urea. Samples were then digested overnight with trypsin (Promega, Madison, WI) at a 1:50 enzyme:substrate ratio, desalted using a C18 Sep-pak cartridge (Waters, UK) and evaporated to dryness before being resuspended in 70% $H_2O$/30% ACN/0.1% TFA and subjected to gel filtration using a Superdex Peptide 10/300 GL column (GE Healthcare). Peak fractions corresponding to crosslinked peptides were then combined and analysed by liquid chromatography tandem mass spectrometry (LC-MS/MS) using an Orbitrap Elite (ThermoScientific, Waltham, MA) as described previously (*Adam et al., 2011*). MZXML data files were analysed using the XQuest software package as described (*Leltner et al., 2014*). Data were represented using Circos with crosslinks weighted by confidence score (generated by XQuest). A confidence score cut-off > 15 was chosen by a combination of FDR < 10% and validation of crosslinks wherever possible using known proximities from crystal structures (lysine α-carbons separated by <27 Å).

## Reconstitution of nucleosomes

Nucleosomes were reconstituted as previously described (*Dyer et al., 2004*). Recombinant *Xenopus* histones were expressed in *E. coli* BL21(DE3) pLysS and purified from inclusion bodies via Sephacryl S-200 gel filtration (GE Healthcare). Stoichiometric amounts of each core histone were incubated together under high salt conditions (2 M NaCl) and the resulting histone octamer purified using a Superdex 200 gel filtration column (GE Healthcare). 216 bp DNA carrying the nucleosome-positioning 601 sequence was PCR amplified from the pGEM-3Z 601 plasmid and purified using a Resource Q anion exchange column (GE Healthcare). Purified DNA, in slight excess to octamers, was mixed together in 2 M NaCl and diluted stepwise with 10 mM Tris (pH 7.5) to reach a final concentration of 100 mM NaCl. The reconstituted nucleosomes were then analysed on a 0.8% Tris-borate agarose gel, and concentrated using a 5000 MWCO spin concentrator (GE Healthcare).

## E3 ligase assays

E3 ligase assays were performed in the presence of reconstituted PRC1 and nucleosomes, with the addition of UBE1 (Boston Biochem), UbcH5c (Boston Biochem), methylated ubiquitin (Boston Biochem) and ATP (Life Technologies). Concentrations of E1, E2 and ubiquitin were pre-optimised such that reactions were pseudo-first order with respect to PRC1. Ubiquitylation reactions were carried out in 50 mM Tris (pH 7.5), 2.5 mM $MgCl_2$ and 0.5 mM DTT. E1, E2, ubiquitin and ATP were pre-incubated for 20 min at 37°C, followed by addition of PRC1 and nucleosomes (0.35 µM final concentration), and then incubated at 37°C for 1 hr. Reactions were quenched with 5 mM EDTA and SDS-PAGE loading buffer (2% SDS, 100 mM Tris-HCl (pH 6.8), 100 mM DTT, 10% glycerol, 1 mg/mL bromophenol blue), boiled for 5 min, and loaded onto SDS-PAGE gels for western blot analysis. Western blots were probed with antibodies specific for histone H2AK119ub1 (*Farcas et al., 2012*) or histone H2A (Millipore 07–146), followed by incubation with LiCOR IRDye secondary antibodies (800CW goat anti-rabbit and goat anti-mouse). Westerns were imaged using the LiCOR Odyssey Fc imaging system, and the fraction of H2AK119ub1 relative to total H2A (non-ubiquitylated H2A plus H2AK119ub1) was quantified. Data were visualised and dose-response curves fitted using GraphPad Prism 6.0.

## Generation of *Rybp^fl/fl^;Yaf2^−/−^* conditional KO embryonic stem cells

*Rybp^fl/fl^* mouse embryonic stem cells, which were previously described and validated (*Hisada et al., 2012*), were grown in DMEM (Gibco, Carlsbad, CA) supplemented with 15% FBS, 10 ng/mL leukemia-inhibitory factor, penicillin/streptomycin, beta-mercaptoethanol, L-glutamine and non-essential amino-acids. Within our laboratory, cells were regularly tested to ensure the absence of mycoplasma contamination. sgRNAs for *Yaf2* (Guide 1: TCTGATCGAGGGGCGACTTT; Guide 2: CGCATGGAACGGCACGGCAC) were cloned into pSpCas9(BB)-2A-Puro (plasmid 48139; Addgene, Cambridge, MA) using a previously described protocol (*Ran et al., 2013*). Cells were transfected with Cas9-sgRNA plasmids using Lipofectamine 3000 (Life Technologies), followed by puromycin

selection (1 µg/mL) for 48 hr. Individual colonies were isolated after approximately 10 days, expanded and screened by PCR for the deletion of *Yaf2* exons 1 and 2. Deletion of *Yaf2* mRNA and protein was confirmed by qRT-PCR and Western Blot, using a YAF2-specific antibody (*Park et al., 2015*). *Rybp*$^{fl/fl}$;*Yaf2*$^{-/-}$ cells were treated with 800 nM 4-hydroxytamoxifen (OHT) for 96 hr to conditionally delete RYBP, which was verified by Western Blot (Millipore AB3637).

## Chromatin immunoprecipitation

Chromatin immunoprecipitation was performed as described previously (*Farcas et al., 2012*), with minor modifications. For non-histone ChIP experiments, cells were fixed for 1 hr in 2 mM EGS, followed by 15 min in 1 % formaldehyde. Reactions were quenched by the addition of glycine to a final concentration of 125 µM. After cell lysis and chromatin extraction, chromatin was sonicated using a BioRuptor sonicator (Diagenode), followed by centrifugation at 16,000 × g for 20 min at 4°C. For histone ChIP experiments we used a native ChIP protocol combined with a calibrated ChIP-seq approach, as previously described (*Bonhoure et al., 2014*; *Hu et al., 2015*; *Orlando et al., 2014*). To achieve this, 1.25 × 10$^7$ *Drosophila melanogaster* S2 cells were spiked in to 5 × 10$^7$ *Rybp*$^{fl/fl}$; *Yaf2*$^{-/-}$ mouse embryonic stem cells with or without OHT treatment. Nuclei were then isolated with RSB buffer (10 mM Tris-HCl (pH 8), 10 mM NaCl, 3 mM MgCl$_2$) supplemented with 0.1 % NP-40 and 5 mM N-ethylmaleimide, followed by MNase digestion for 5 min with 150 U MNase (Fermentas, Waltham, MA) in 1 ml RSB supplemented with 0.25 M sucrose, 3 mM CaCl$_2$ and 10 mM N-ethylmaleimide. Digestions were stopped with EDTA, before nuclei were pelleted by centrifugation at 1500 x g and the soluble S1 fraction collected. Pelleted nuclei were then resuspended in 300 µl nucleosome release buffer (10 mM Tris-HCl (pH 7.5), 10 mM NaCl, 0.2 mM EDTA, 10 mM N-ethylmaleimide), incubated at 4°C for 1 hr with gentle rotation, and then gently passed through a 27G syringe needle five times. Following centrifugation to pellet the insoluble material, the soluble S2 fraction was collected and combined with the S1 fraction.

Immunoprecipitations were performed overnight at 4°C using chromatin corresponding to 5 × 10$^6$ cells. For non-histone ChIP experiments, chromatin was diluted in ChIP dilution buffer (1% Triton-X100, 1 mM EDTA, 20 mM Tris-HCl (pH 8), 150 mM NaCl), while for histone ChIPs, combined S1/S2 fractions were diluted in native ChIP incubation buffer (70 mM NaCl, 10 mM Tris (pH 7.5), 2 mM MgCl$_2$, 2 mM EDTA, 0.1% Triton). Immunoprecipitations were carried out in a total volume of 1 ml using approximately 3 µg of antibody. Antibodies used for ChIP experiments were anti-H2AK119ub1 (Cell Signalling D27C4), anti-H3K27me3 (generated in-house), anti-RYBP (Millipore AB3637), anti-RING1B (Cell Signalling D22F2) and anti-SUZ12 (Cell Signalling D39F6). Antibody-bound chromatin was isolated on protein A agarose beads (RepliGen, Waltham, CA) or protein A magnetic Dynabeads (Invitrogen, Carlsbad, CA). For non-histone ChIPs, washes were performed with low salt buffer (0.1% SDS, 1% Triton, 2 mM EDTA, 20 mM Tris-HCl (pH 8.1), 150 mM NaCl), high salt buffer (0.1% SDS, 1% Triton, 2 mM EDTA, 20 mM Tris-HCl (pH 8.1), 500 mM NaCl), LiCl buffer (0.25 M LiCl, 1% NP40, 1% sodium deoxycholate, 1 mM EDTA, 10 mM Tris-HCl (pH 8.1)) and TE buffer (x2 washes) (10 mM Tris-HCl (pH 8.0), 1 mM EDTA). Native histone ChIPs were instead washed four times with native ChIP wash buffer (20 mM Tris-HCl (pH 7.5), 2 mM EDTA, 125 mM NaCl and 0.1% Triton), followed by a final TE buffer wash. To prepare ChIP-seq material, ChIP DNA was eluted, and cross-links reversed at 65°C in the presence of 200 mM NaCl where required. Samples were then treated with RNase and proteinase K before being purified with ChIP DNA Clean and Concentrator kit (Zymo, Irvine, CA). For calibrated ChIP-seq experiments, an input control was prepared for each individual sample, to allow the quantitation of spike-in consistency.

## Massively parallel sequencing, data processing and normalisation

ChIP-seq libraries were prepared using the NEBNext Ultra DNA Library Prep Kit, and sequenced as 40 bp paired-end reads on Illumina NextSeq500 platform. All ChIP-seq experiments were carried out in biological triplicate. To analyse the ChIP-seq experiments, reads were aligned to the mouse mm10 genome using bowtie2 (*Langmead and Salzberg, 2012*) with the '-no-mixed' and '-no-discordant' options, and non-uniquely mapping reads were discarded. ChIP-seq experiments which contained a spike-in genome (*D. melanogaster*) were aligned against a concatenated genome of the two genome sequences (mm10+dm6), and reads which mapped more than once were discarded. PCR duplicates were removed using SAMtools (*Li et al., 2009*).

Spiking an identical concentration of *D. melanogaster* cells into our quantitative ChIP-seq experiments allows for calibration of each sample for IP efficiency and total mouse cell number. When comparing ChIP-seq for untreated and OHT-treated cell lines using an internal calibration, the number of mm10 reads were randomly downsampled to reflect the total number of dm6 reads in that sample. Furthermore, to adjust for variation in the dm6/mm10 ratio between biological replicates, each sample was adjusted using the percentage of dm6 reads relative to mm10 reads in each sample's input DNA. Biological replicates for ChIP-seq samples without internal spike-in were randomly downsampled to contain the same number of reads for each individual replicate. Genome coverage tracks were made using the pileup function of MACS2 (*Zhang et al., 2008*) and visualised using the UCSC genome browser.

## Peak calling, read count quantitation and analysis

We identified regions of H2AK119ub1 enrichment using the dpeak function of DANPOS2 (-q 40 –kw 750 –kd 1500, height_logP > 100) (*Chen et al., 2013*) and peaks closer than 5 kilobases were merged. RING1B, SUZ12 and H3K27me3 peaksets were generated using MACS2 (-broad), and peaks closer than 2 kilobases were merged. Read counts within intervals were generated using custom script GFF.ReadCountFromList.pl. For non-calibrated ChIP-seq, reads per kilobase per million (RPKM) were calculated, while for calibrated ChIP-seq, normalised reads per kilobase (NRPK) were calculated (after normalisation to calibration genome). For each interval, biological triplicate NRPK or RPKM values were compared between untreated and OHT-treated samples using a two-sample T-Test, with a *p* value threshold of <0.05 being used to identify significant difference. Log2 fold change values were visualised using R (v 3.2.1), with scatterplots coloured by density using stat_density2d. Regression and correlation analyses were also performed in R using standard linear models and Pearson correlation respectively. Metaplots were made using HOMER (version 2) (*Heinz et al., 2010*), Heatmaps and genome-wide correlation analyses were generated using deepTools2.

## Identification of variant-enriched and shared PRC1 target sites

A ±10 kb window centred on H2AK119ub1 intervals was provided to deepTools2 for *k*-means clustering based on RYBP (this study) and CBX7 (*Morey et al., 2013*) ChIP-seq (*Ryan et al., 2016*). This generated two clusters, variant-enriched (RYBP-enriched but CBX7-depleted) and shared (RYBP- and CBX7-enriched).

## Nuclear RNA-seq sample generation and analysis

$1 \times 10^7$ *Rybp*$^{fl/fl}$;*Yaf2*$^{-/-}$ cells with or without OHT were resuspended in 1 mL HS Lysis buffer (50 mM KCl, 10 mM MgSO$_4$.7H$_2$0, 5 mM HEPES, 0.05% NP40 (IGEPAL CA630)), 1 mM PMSF, 3 mM DTT) for 1 min at room temperature to isolate nuclei. Nuclei were centrifuged at 1000 × *g* for 5 min at 4°C, followed by a total of three washes with ice-cold RSB buffer. Nuclei were then resuspended in 1 mL of TriZOL reagent (ThermoScientific) according to the manufacturer's protocol. Nuclear RNA was treated with the TURBO DNA-free Kit (ThermoScientific) and depleted for rRNA using the Ribo-Minus kit and protocol (ThermoScientific). RNA-seq libraries were prepared using the NEBNext Ultra Directional RNA-seq kit (NEB) and libraries were sequenced on the Illumina NextSeq500 with 40 bp paired-end reads.

RNA-seq reads were initially aligned against the rRNA genomic sequence (GenBank: BK000964.3) using bowtie2 to filter out rRNA fragments, prior to alignment against the mm10 genome using the STAR RNA-seq aligner (*Dobin et al., 2013*). To improve mapping of nascent, intronic sequences, reads which failed to map using STAR were aligned against the genome using bowtie2. Biological triplicate read counts were determined using a custom script (GFF_PairedEndRNA.ReadCount.pl) for a custom-built, non-redundant mm10 gene set. Briefly, mm10 refGene genes were filtered on size (>200 bp), gene body and TSS mappability, unique TSS and TTS, in order to remove poorly mappable and highly similar transcripts. The final set of 20,633 genes were used for differential analysis using DESeq2 (*Love et al., 2014*), with default settings and a significance threshold of *padj* < 0.05. Previously published nuclear RNA-seq from *RING1B*$^{fl/fl}$;*RING1A*$^{-/-}$ was aligned and processed in an identical manner (*Schoenfelder et al., 2015*). Statistical significance between Polycomb target TSS and non-Polycomb TSS was assessed using Wilcoxon signed rank test. Venn diagram overlap of significantly increased genes was performed using BioVenn (*Hulsen et al., 2008*).

## Accession numbers

ChIP-seq and RNA-seq data from the present study are available for download at GSE83135. Previously published datasets used for analysis include wildtype embryonic stem cell (E14) H3K4me3 from the ENCODE project (GSE31039), CBX7 ChIP-seq (GSE42466; *Morey et al., 2013*), RNA polymerase II ChIP-seq (GSE34520; *Brookes et al., 2012*), MNase-seq (GSE59064; *Anderson and Slotkin, 1975*); RING1A$^{-/-}$;RING1B$^{fl/fl}$ nuclear RNA-seq (E-MTAB-3125; *Schoenfelder et al., 2015*) and RING1B$^{I53A}$ mutant (GSE69978; *Illingworth et al., 2015*).

## Acknowledgements

Work in the Klose lab is supported by the Wellcome Trust, the Lister Institute of Preventive Medicine, and Exeter College University of Oxford, EMBO, and the European Research Council. NRR acknowledges a Junior Research Fellowship from St John's College, University of Oxford. BMK was supported by the John Fell Fund 133/075 and the Wellcome Trust grant 097813/Z/11/Z, and RF by a Kennedy Trust Fund awarded to BMK. We would like to thank Haruhiko Koseki and Miguel Vidal for the kind gift of *Rybp*$^{fl/fl}$ mouse embryonic stem cells, in addition to Imre Berger for supplying us with the multiBAC expression system and HoGeun Yoon for the YAF2-specific antibody. We would also like to acknowledge Tom Owen-Hughes and the late Jonathan Widom for providing us with the pGEM-3Z 601 plasmid, and Huseyin Besir and Ines Racke at the Protein Expression and Purification Core Facility, EMBL for Baculovirus expression. We thank Rebecca Konietzny and Philip Charles for help with mass spectrometry analysis and data processing, and Tom Milne and Emilia Dimitrova for critical reading of the manuscript.

## Additional information

### Funding

| Funder | Grant reference number | Author |
| --- | --- | --- |
| European Research Council | Consolidator grant, 681440 | Robert J Klose |
| Wellcome Trust | Senior Research Fellowship, 098024/Z/11/Z | Robert J Klose |
| Lister Institute of Preventive Medicine | | Robert J Klose |
| Wellcome Trust | 097813/Z/11/Z | Benedikt M Kessler |
| John Fell Fund, University of Oxford | 133/075 | Benedikt M Kessler |
| Kennedy Memorial Trust | | Benedikt M Kessler |

The funders had no role in study design, data collection and interpretation, or the decision to submit the work for publication.

### Author contributions

NRR, HWK, NPB, Conception and design, Acquisition of data, Analysis and interpretation of data, Drafting or revising the article; NAF, Analysis and interpretation of data, Drafting or revising the article; KJIE, Acquisition of data, Analysis and interpretation of data; RF, BMK, Acquisition of data, Analysis and interpretation of data, Drafting or revising the article; RJK, Conception and design, Analysis and interpretation of data, Drafting or revising the article

### Author ORCIDs

Roman Fischer, http://orcid.org/0000-0002-9715-5951
Robert J Klose, http://orcid.org/0000-0002-8726-7888

## Additional files

### Major datasets

The following dataset was generated:

| Author(s) | Year | Dataset title | Dataset URL | Database, license, and accessibility information |
|---|---|---|---|---|
| Rose NR, King HW, Blackledge NP, Ember K, Fischer R, Kessler BM, Klose RJ | 2016 | RBYP stimulates PRC1 to shape chromatin-based communication between polycomb repressive complexes | https://www.ncbi.nlm.nih.gov/geo/query/acc.cgi?acc=GSE83135 | Publicly available at the NCBI Gene Expression Omnibus (accession no: GSE83135). |

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
