## [Decision Letter]

Thank you for submitting your article "RYBP stimulates PRC1 to shape chromatin-based communication between Polycomb repressive complexes" for consideration by *eLife*. Your article has been reviewed by three peer reviewers, one of whom is a member of our Board of Reviewing Editors and the evaluation has been overseen by Jessica Tyler as the Senior Editor. The following individual involved in review of your submission has agreed to reveal her identity: Wendy A Bickmore (Reviewer #3).

The reviewers have discussed the reviews with one another and the Reviewing Editor has drafted this decision to help you prepare a revised submission.

Summary:

In this manuscript, Rose and colleagues use biochemical and genetic approaches to address how the RYBP subunit of non-canonical PRC1 (ncPRC1) complexes contribute to transcriptional regulation and if they also contribute to PRC2 complex activity. This is in the context of recent studies reporting that the ubiquitination of H2A (mediated by ncPRC1 complexes) is both capable of recruiting the PRC2 complex to an artificial gene locus (Blackledge et al., 2014, Cell; Cooper et al., 2014) and increasing the activity of PRC2 in vitro (Kalb et al., 2014, Nature Structural Molecular Biology). However, more recent genetic evidence in Ring1a/b double knockout cells have shown that a complete ablation of ncPRC1 activity did not affect global H3K27me3 levels in gut tissues (Chiacchiera et al., Cell Stem Cell, 2016). Furthermore, while the loss of PRC2 mediated H3K27me3 leads to HOX gene derepression in *Drosophila* (Pengelly et al., Science, 2013), the complete loss of H2A ubiquitination did not phenocopy this (Pengelly et al., Genes & Development 2015). Likewise, recent genetic studies questioned the importance of H2A ubiquitination for Polycomb silencing in mice (Illingworth et al., Genes Dev. 2015). Despite this, we know from these studies and others, that ncPRC1 and indeed H2Aub are essential for embryonic development. Finally, that RYBP can stimulate H2Aub1 by a subset of PRC1 complexes is already established in ES cells and in flies (Gao et al., 2012; Fereres et al., 2014, PLoS One). Moreover, cross-talk between PRC1/H2Aub1 and PRC2/H3K27me3 has been examined previously (Tavares et al., 2012; Blackledge et al., 2014; Cooper et al., 2014; Kalb et al., 2014; Illingworth et al., Genes Dev. 2015; Pengelly et al., Genes Dev 2015; Fereres et al., 2014, PLoS One; Morey et al., Cell Reports 3, 2013), but these studies resulted in rather diverse conclusions.

In this overall well-done study, the authors firstly establish an in vitro ubiquitination assay for ncPRC1, and use it to explore the role of the RYBP subunit (Figure 1–Figure 4). They discover that RYBP is capable of considerably boosting H2A ubiquitination in vitro. In Figure 5–Figure 9, they use knockout mouse embryonic stem (ES) cells to explore the roles of RYBP in vivo. They compare a Yaf2 knockout ES cells and Rybp/Yaf2 double knockout ES cells to examine the effect on the global levels of H2AK119ub1 (Western blot) and at selected Polycomb target genes (ChIP-seq). While surprisingly, they observe that the overall global levels of H2AK119ub1 are apparently unchanged in the RYBP/Yaf2 double knockout ES cells, they detect moderate local reductions of H2AK119ub1 at sites normally occupied by Rybp and Ring1b. Taken together with their in vitro observations, these results suggest that Rybp and Yaf2 are not essential for the H2AK119ub1 activity of ncPRC1 complexes and rather are co-factors that can augment activity, perhaps at more weakly bound sites. They then go on to correlate local reductions in H2AK119ub1 with moderate reductions in H3K27me3 and slight increases in transcription at some of the affected Polycomb target genes. However, they report that this reduced H3K27me3 does not correlate with reduced PRC2 complex occupancy, as measured by ChIP-seq analysis of Suz12 in their Rybp/Yaf2 double knockout ES cells. The authors use these observations to propose a model for the communication between PRC1 and PRC2, at least on a number of gene regulatory loci.

Whereas the reviewers found the manuscript potentially interesting for the readership of *eLife*, because it shed light on the function of RYBP and the communication between PRC1 and PRC2, several major concerns were raised that will need to be addressed experimentally.

Essential revisions:

1) The authors should avoid making quantitative conclusions about Ring1b (Figure 6) and Suz12 (Figure 8) in the wild-type versus Rybp/Yaf2 double knockout ES cells. They rightly point out that the ChIP-RX method (Orlando et al., Cell Reports, 2014,) is the correct approach to quantify potential changes in the H2AK119ub1 and H3K27me3 histone marks. However, they employ non-quantitative ChIP-seq to look at the Ring1b and Suz12 binding profiles. They should instead employ a careful, quantitative ChIP-qPCR approach, checking multiple representative sites, in order to properly evaluate if Ring1B and Suz12 occupancy change in the Rybp/Yaf2 double knockout ES cells (Figure 6 and Figure 8). This is especially important for both their model that RYBP primarily influences ncPRC1 activity (and not level, or localization of the complex, as published by others see e.g. Gao et al., 2012; Tavares et al., 2012; Morey et al., Cell Reports 3, 2013) and the open question whether the presence of H2AK119ub1 might either lead to increased recruitment of PRC2 or increase its activity (again, the authors should content with the already published work). Based on ChIP data the authors conclude that H2A ubiquitination is required for normal H3K27 methylation and proposed an "activity-based communication between PRC1 and PRC2". While this conclusion is likely to be correct (this point has basically been made in earlier publications), this cannot be concluded solely from ChIP data in this manuscript. In fact, Kalb et al. (2014) already did an experiment that allows such a mechanistic interpretation by showing that nucleosomes containing H2Aub1 are a better substrate for H3K27 methylation by PRC2.

2) The authors should carry out gene ontology (GO) analysis of the genes that lose both H2AK119ub1 and H3K27me3 at their promoters in the Rybp/Yaf2 double knockout ES cells in a bid to identify ncPRC1 specific sets of target genes and compare to similar work in previous studies. It would be interesting to further delineate those H3K27me3 positive genes that do and don't have reduced H3K27me3 in the Rybp/Yaf2 double knockout ES cells. This would be important for the readers in light of previous work delineating the biological roles of cPRC1 and ncPRC1 (Pengelley et al., 2015 and Illingworth et al., 2015). These studies found that H2A ubiquitination appears to be dispensable for Polycomb dependent repression of the Hox genes. However, since H2Aub is required for both fly and mouse viability, it becomes interesting to begin to determine which genes are dependent on ncPRC1 mediated H2AK119ub1 for their regulation.

3) To better report the genome-wide binding profiles of cPRC1 and ncPRC1 in ES cells, the authors should include heat maps of their ChIP-seq data for all transcription start sites (+/- 5kb) for Cbx7 (Morey et al., 2013, Cell Reports), Ring1b (+/-OHT), Rybp (+/-OHT), H2AK119ub1 (+/-OHT) and H3K27me3 (+/-OHT). This would help delineate the cPRC1 and ncPRC1 unique target genes and it may shed light on whether loss of Rybp affects H3K27me3 levels at shared ncPRC1 and cPRC1 target genes or on ncPRC1 specific target genes only.

4) Following this heat map analysis, the authors should investigate whether the 230 Polycomb target genes that have an increase in transcription in their Rybp/Yaf2 double knockout cells (Figure 9) are ncPRC1 specific, cPRC1 specific or ncPRC1 and cPRC1 shared target genes. They should provide ChIP-qPCRs of Suz12, H3K27me3 and Ring1b at a representative set of these gene promoters.

5) In Figure 1–Figure 4 reconstituted histone H2A ubiquitination assays are used in which different recombinant PRC1 assemblages are compared by titrating increasing amounts of enzyme with a fixed substrate and the% conversion at the end of the reaction is determined. The correct way to compare enzymatic activities is to determine reaction rates. I.e. measure conversion over time using a fixed amount of enzyme and substrate! This experiment should be included for a few key complexes. Another critical issue is that the ratio enzyme vs substrate (nucleosomes) is not clear. It is essential that the substrate is in excess over the enzyme (this should be shown). Finally, a better analysis of the nucleosomes used in the assays need to be provided. E.g. are there free histones in the prep? An additional nucleosome purification step (e.g. glycerol centrifugation or size exclusion chromatography) would give more confidence. This is also important because the results in this manuscript are at odds with published work from other groups (e.g. Gao et al. 2012; Tavares et al., 2012). Finally, subsection “PCGF1-PRC1 is a highly modular protein complex” – and Figure 1. It appears that the absence of RYBP leads to consistently reduced levels of PCGF1 (lanes 4 to 6) and possibly a slight reduction in BCOR (lane 5). This suggests that RYBP may help to stabilize these interactions. Quantitation of this blot should be performed and these result discussed in the text.

6) The Introduction, and actually most of the manuscript, does not give a clear overview of what has already been published on this topic. It is not that many major references were missing (although e.g. the work of the Sixma lab (Buchwald et al., 2006) should be included), rather the Introduction does not clearly summarized the work already done. In addition, somewhat more attention to work in the fly would be helpful to the reader as well. E.g. ncPRC1 complexes and there role in histone ubiquitylation were first described in the fly (Lagarou et al., 2008). Finally, in the Results and DDiscussion sections, findings by others with similar experiments should be mentioned and where appropriated differences should be discussed. The whole manuscript strongly emphasizes the catalytic functions of PRC1/2 but this needs to be much better balanced by a discussion of the substantial evidence – both in vitro and in vivo, and in flies and mammals – that PRC1 executes many of its functions through a structural change in the underlying chromatin and that is independent of its catalytic activity.

---

## [Author Response]

[…]

*Essential revisions:*

*1) The authors should avoid making quantitative conclusions about Ring1b (Figure 6) and Suz12 (Figure 8) in the wild-type versus Rybp/Yaf2 double knockout ES cells. They rightly point out that the ChIP-RX method (Orlando et al., Cell Reports, 2014,) is the correct approach to quantify potential changes in the H2AK119ub1 and H3K27me3 histone marks. However, they employ non-quantitative ChIP-seq to look at the Ring1b and Suz12 binding profiles. They should instead employ a careful, quantitative ChIP-qPCR approach, checking multiple representative sites, in order to properly evaluate if Ring1B and Suz12 occupancy change in the Rybp/Yaf2 double knockout ES cells (Figure 6 and Figure 8). This is especially important for both their model that RYBP primarily influences ncPRC1 activity (and not level, or localization of the complex, as published by others see e.g. Gao et al., 2012; Tavares et al., 2012; Morey et al., Cell Reports 3, 2013) and the open question whether the presence of H2AK119ub1 might either lead to increased recruitment of PRC2 or increase its activity (again, the authors should content with the already published work).*

We agree with the reviewer(s) that this is an important point and we have now performed new ChIP-qPCR analysis for RING1B, SUZ12, H2AK119ub1, and H3K27me3 in biological triplicate at multiple representative sites. This analysis has confirmed that RING1B and SUZ12 are largely retained, or even in some instances elevated in the case of SUZ12 (consistent with our ChIP-seq meta-analysis for SUZ12 – see Figure 8), while H2AK119ub1 and H3K27me3 are reduced following removal of RYBP/YAF2 as suggested by our original ChIP-seq analysis. These new ChIP-qPCR results have been integrated as new supplementary figures (Figure 6—figure supplement 1 and Figure 8—figure supplement 2), and further support the argument that RYBP/YAF2 acts to influence PRC1 and PRC2 activity rather than their binding to chromatin.

*Based on ChIP data the authors conclude that H2A ubiquitination is required for normal H3K27 methylation and proposed an "activity-based communication between PRC1 and PRC2". While this conclusion is likely to be correct (this point has basically been made in earlier publications), this cannot be concluded solely from ChIP data in this manuscript. In fact, Kalb et al. (2014) already did an experiment that allows such a mechanistic interpretation by showing that nucleosomes containing H2Aub1 are a better substrate for H3K27 methylation by PRC2.*

The reviewer(s) are correct in pointing out that our in vivo ChIP-seq analysis and the ‘activity-based communication’ model we arrive at is in agreement with, and strengthened by, the elegant in vitro work performed by Kalb et al. (2014) in which they demonstrated that the activity of PRC2 was stimulated by H2AK119ub1. Our new observations provide clear genome-wide evidence that reduced H2AK119ub1 at PcG target sites in vivo predominantly influences PRC2 activity and not target site binding. We have included the following statement to clarify this point in subsection “RYBP-dependent H2AK119ub1 shapes PRC2 activity”:

“These important observations demonstrate that RYBP-dependent H2AK119ub1 shapes the activity of PRC2 in depositing H3K27me3, and suggests that the observed stimulation of PRC2 by nucleosomes containing H2A mono-ubiquitylation in vitro is also relevant in vivo (Kalb et al., 2014).”

*2) The authors should carry out gene ontology (GO) analysis of the genes that lose both H2AK119ub1 and H3K27me3 at their promoters in the Rybp/Yaf2 double knockout ES cells in a bid to identify ncPRC1 specific sets of target genes and compare to similar work in previous studies. It would be interesting to further delineate those H3K27me3 positive genes that do and don't have reduced H3K27me3 in the Rybp/Yaf2 double knockout ES cells. This would be important for the readers in light of previous work delineating the biological roles of cPRC1 and ncPRC1 (Pengelley et al., 2015 and Illingworth et al., 2015). These studies found that H2A ubiquitination appears to be dispensable for Polycomb dependent repression of the Hox genes. However, since H2Aub is required for both fly and mouse viability, it becomes interesting to begin to determine which genes are dependent on ncPRC1 mediated H2AK119ub1 for their regulation.*

We agree this is an interesting and important question and thank the reviewer(s) for the suggestion. To examine these relationships in more detail we undertook two complementary new analyses.

Firstly, as suggested by the reviewer(s), we identified polycomb targets which relied the most (RYBP-dependent) or the least (RYBP-independent) on RYBP for H3K27me3, and compared their gene ontology (GO) enrichment terms (Figure 10). Both groups of polycomb targets were enriched for GO terms implicated in development (e.g. “system development”, “developmental process”, “organ development” (Figure 10)), consistent with polycomb complexes playing an important role in regulating developmental genes in embryonic stem cells. However, interestingly, sites which retained the most H3K27me3 in the absence of RYBP/YAF2 exhibited enrichment for DNA binding and sequence-specific transcription factors (Autho response Figure 1). These transcription factors are often master regulators involved in developmental decision making processes, and are found in large and very CpG dense islands that have high levels of polycomb protein occupancy, H3K27me3, and H2AK119ub1 ((Long et al., 2013) and Figure 10). In this context the stimulatory activity of RYBP and H2AK119ub1 appears less pronounced, suggesting that these sites may have sufficient PRC2 occupancy to render RYBP and H2AK119ub1-dependent stimulation less relevant.

Author response image 1.GO analysis of RYBP-dependent and –independent H3K27me3 genes.(**A**,**B**) PRC1 target genes with the largest reductions in H3K27me3 (RYBP-dependent H3K27me3; Top 25%), or the PRC1 targets with the smallest reductions in H3K27me3 (RYBP-independent) were used for gene ontology analysis using HOMER. (**C**) Metaplot analysis at PRC1 target sites with RYBP-dependent or RYBP-independent H3K27me3. This reveals that RYBP-independent H3K27me3 is associated with the largest polycomb domains, as measured by RING1B, SUZ12 and CpG density.**DOI:**
http://dx.doi.org/10.7554/eLife.18591.024

Secondly, to further investigate the relationship between canonical PRC1 (cPRC1) and variant (vPRC1), we performed de novo clustering based on enrichment of RYBP (this study) and CBX7 (Morey et al., 2013) at H2AK119ub1 sites and arrived at a shared and variant-enriched set of PcG target sites (see essential reviewer point 3 for more details). In agreement with the segregation of sites based on changes in H3K27me3, both groups were strongly associated with developmental processes; however the shared sites exhibited a slight preference for sequence-specific transcription factors. Variant enriched sites also encompassed transcription factors, but in general represented a set of genes that included a more broad range of biological processes.

Together this suggests that sequence-specific transcription factors may have evolved to use a more robust complement of cPRC1 and vPRC1 activities to help maintain H2AK119ub1 while vPRC1 may play a broader role in shaping H2AK119ub1 associated with a range of biological processes. We have integrated these analyses into the revised manuscript, with GO term analysis included in Figure 7—figure supplement 1. We have also discussed these findings where relevant (see subsection “RYBP-dependent stimulation is essential for H2AK119ub1 at sites with low PRC1 occupancy”).

*3) To better report the genome-wide binding profiles of cPRC1 and ncPRC1 in ES cells, the authors should include heat maps of their ChIP-seq data for all transcription start sites (+/- 5kb) for Cbx7 (Morey et al., 2013, Cell Reports), Ring1b (+/-OHT), Rybp (+/-OHT), H2AK119ub1 (+/-OHT) and H3K27me3 (+/-OHT). This would help delineate the cPRC1 and ncPRC1 unique target genes and it may shed light on whether loss of Rybp affects H3K27me3 levels at shared ncPRC1 and cPRC1 target genes or on ncPRC1 specific target genes only.*

We thank for the reviewer(s) for this useful suggestion, and agree that the manuscript would benefit from a better understanding of the relative contribution of RYBP towards previously proposed canonical and/or variant PRC1 targets (as also covered in essential reviewer point 2). To address this important point, we implemented the reviewer(s) suggestion and profiled H2AK119ub1, RING1B, RYBP, and SUZ12 in the presence and absence of RYBP/YAF2 at TSS ± 5kb clustered based on the presence of RING1B and the relative levels of RYBP (vPRC1) and CBX7 (cPRC1). This revealed a set of “shared” sites, with high levels of vPRC1 and cPRC1, and “variant-enriched” sites, which exhibited lower levels of CBX7 (Figure 11). In contrast to previous work (Morey et al., 2013) we failed to identify cPRC1 (CBX7)-enriched sites that lack vPRC1 (RYBP). Initially this was puzzling to us, but a closer inspection of the genes that were previously reported to lack RYBP yet have CBX7 binding ((Morey et al., 2013) Figure 2) revealed a clear enrichment for RYBP in our experiments and this signal was lost in response to tamoxifen-induced removal of RBYP (Figure 12). The differences in RYBP ChIP enrichment between our studies is presumably the result of differing ChIP protocols (e.g. formaldehyde only crosslinking (Morey et al) versus formaldehyde/EGS crosslinking (our study)). Nevertheless, it is clear that CBX7 occupied sites fall into the shared category that are also occupied by RYBP. In the revised manuscript, we now highlight this important point (subsection “RYBP-dependent stimulation is essential for H2AK119ub1 at sites with low PRC1 occupancy”). Intriguingly, we observed that both shared and variant-enriched PcG TSS appeared to be similarly affected in their reductions of H2AK119ub1 and H3K27me3 (Figure 11), and we did not observe any obvious changes in RING1B or SUZ12 occupancy at either group. However, a visual examination of polycomb domains and their associated genes indicated that the use of a simple fixed window surrounding the TSS would not likely capture the behaviour of the entire domain, particularly when considering H2AK119ub1 and H3K27me3 which are often broad in nature (Figure 11). Therefore, we instead opted to use a clustering and heat mapping approach focused on H2AK119ub1 intervals. This revealed that although all polycomb target sites experience reductions in H2AK119ub1 and H3K27me3, variant-enriched sites showed a slightly larger reduction in both of these modifications (Figure 7, Figure 8). We have now incorporated these analyses and interpretations into the revised Figures and manuscript highlighting the preferential contribution of RYBP towards vPRC1-enriched sites and described these observations on in the revised manuscript.

Author response image 2.TSS clustering and heatmap analysis of variant and canonical PRC1.(**A**) Non-redundant mm10 refGene TSSs (n=20634) were clustered into Polycomb (PcG) or non-PcG TSS. These were then further clustered based on RYBP and CBX7 binding (Morey et al., 2013) into shared (RYBP and CBX7 aka variant and canonical PRC1) and variant-enriched (RYBP-enriched) TSSs. Non-PcG TSSs were clustered by H3K4me3 to identify in active and inactive TSS. (**B**) Genomic snapshot of *Six1* and *Shh* TSSs. A fixed window of ± 5kb is depicted from the TSS (arrow). This simple windowing suffers from the fact that it does not capture the true polycomb chromatin domain. In contrast intervals based on H2AK119ub1 more accurately represent the region corresponding to the polycomb chromatin domain.**DOI:**
http://dx.doi.org/10.7554/eLife.18591.025

Author response image 3.Analysis of RYBP occupancy at previously described canonical PRC1-specific sites.Genomic snapshots of ChIP-seq signal for RYBP (datasets from this study and (Morey et al., 2013)), CBX7 (Morey et al., 2013), RING1B (this study) and SUZ12 (this study) at three genes previously reported as being occupied by cPRC1 and RYBP-free/depleted (Morey et al., 2013). Our RYBP ChIP-seq data (complete with RYBP knockout control (OHT)), clearly shows enrichment for RYBP at these genes. Importantly, RYBP enrichment at these sites is comparable in magnitude to other ncPRC1-enriched and shared sites (for example compare with Figure 7).**DOI:**
http://dx.doi.org/10.7554/eLife.18591.026

*4) Following this heat map analysis, the authors should investigate whether the 230 Polycomb target genes that have an increase in transcription in their Rybp/Yaf2 double knockout cells (Figure 9) are ncPRC1 specific, cPRC1 specific or ncPRC1 and cPRC1 shared target genes.*

To address this question, we segregated polycomb-occupied TSS into shared and variant-enriched PRC1 targets and examined which of these two sets the reactivated genes fell into. This analysis showed that upregulated genes fell roughly into both categories. We have now indicated this in the text of the revised manuscript in subsection “Loss of RYBP culminates in gene reactivation at sites where Polycomb chromatin domains are compromised” and Figure 9—figure supplement 2.

*They should provide ChIP-qPCRs of Suz12, H3K27me3 and Ring1b at a representative set of these gene promoters.*

As requested, we have performed new ChIP-qPCRs for RING1B, SUZ12, H3K27me3 and H2AK119ub1 in biological triplicate at a set of representative genes that are up-regulated upon loss of RYBP/YAF2. The results of these analyses further support the contention (Figure 9) that up-regulated genes tended to have compromised polycomb domains, as evidenced by reductions in RING1B and SUZ12 binding (Figure 9—figure supplement 3).

*5) In Figure 1–Figure 4 reconstituted histone H2A ubiquitination assays are used in which different recombinant PRC1 assemblages are compared by titrating increasing amounts of enzyme with a fixed substrate and the% conversion at the end of the reaction is determined. The correct way to compare enzymatic activities is to determine reaction rates. I.e. measure conversion over time using a fixed amount of enzyme and substrate! This experiment should be included for a few key complexes.*

The reviewer is correct in noting this experimental setup does not conform to Michaelis-Menten analysis of reaction rates. This was a deliberate decision that we made in designing these experiments and we will explain why in more detail here. Currently, the only method we have found to easily measure PRC1 activity in vitro is to follow the conversion of H2A to ubiquitylated H2A by western blot. As indicated by the reviewer, to set up an experiment to measure classical Michaelis-Menten kinetics, one must (a) have substrate in large excess over enzyme, and (b) vary substrate amount relative to fixed enzyme concentration. In the available experimental set-up, western blot analysis has not proven sensitive enough to robustly detect conversion of H2A to uH2A at low substrate concentrations, effectively excluding Michaelis-Menten analysis. Furthermore, the analysis of Michaelis-Menten kinetics in the multi-reaction ubiquitylation cascade (E1, E2, and E3) necessary to achieve uH2A would be extremely challenging to design and to interpret correctly. To our knowledge this has not previously been achieved for histone ubiquitylation reaction cascades. Therefore we chose to employ a dose-response approach in order to measure and compare the relative E3 ubiquitin ligase activity of different PRC1 complex assemblies by following percentage conversion of a fixed amount of substrate. In fact, similar approaches have previously been used to compare the relative activities of PRC1 complexes (Gao et al., 2012, Tavares et al., 2012, Buchwald et al., 2006, Cao et al., 2005, Li et al., 2006), enabling us to broadly compare our new findings with previous work in this area. Nevertheless, to avoid any confusion we have now clarified this point in the text of manuscript and explicitly indicate that we are comparing relative activities as opposed to reaction rates in the Results section.

*Another critical issue is that the ratio enzyme vs substrate (nucleosomes) is not clear. It is essential that the substrate is in excess over the enzyme (this should be shown).*

As noted above, this point is particularly relevant in situations where Michaelis-Menten-type kinetic analyses are being conducted, which we have not attempted to do here. However, we realise that in the initial submission the concentration of nucleosome in ubiquitylation reactions was inadvertently omitted. We have now added this information to the Materials and methods in the revised manuscript.

Finally, a better analysis of the nucleosomes used in the assays need to be provided. E.g. are there free histones in the prep? An additional nucleosome purification step (e.g. glycerol centrifugation or size exclusion chromatography) would give more confidence. This is also important because the results in this manuscript are at odds with published work from other groups (e.g. Gao et al. 2012; Tavares et al., 2012).

Octamers were purified by gel filtration and displayed the expected equimolar ratio of histones. In the nucleosome reconstitutions, DNA was always in slight excess over octamer, and tris-borate native agarose gels used to examine the reconstituted nucleosomes indicated that only a small proportion of unincorporated DNA was present relative to nucleosomal DNA. This suggests a near complete incorporation of octamers into nucleosomes under our reconstitution conditions as we have previously published (Zhou et al., 2012). Furthermore, it has been demonstrated that PRC1 is highly specific towards H2A in the context of nucleosomes, and any non-specific activity on free histone results in polyubiquitylation (Elderkin et al., 2007), which would not be detected in our western blotting measurements of H2AK119ub1. Finally we do not believe that the results of our in vitro assays are at odds with either of the studies mentioned by the reviewer. In Gao et al., it was also demonstrated that RYBP stimulates the activity of a PCGF4-containing complex (see (Gao et al., 2012) Figure 5), which is in agreement with our observations in Figure 3. In Tavares et al., the influence of RYBP on catalytic activity of PRC1 was tested using a PCGF2-containing complex, which we did not directly examine in our study.

*Finally, subsection “PCGF1-PRC1 is a highly modular protein complex” and Figure 1. It appears that the absence of RYBP leads to consistently reduced levels of PCGF1 (lanes 4 to 6) and possibly a slight reduction in BCOR (lane 5). This suggests that RYBP may help to stabilize these interactions. Quantitation of this blot should be performed and these result discussed in the text.*

We have now quantified PCGF1 in the indicated complex preparations and find the levels are reduced in the absence of RYBP as suggested by the reviewer. We thank the reviewer for pointing this out as we believe it further supports the observations from crosslinking mass spectrometry analysis (Figure 4) that RYBP inclusion in the complex may lead to structural alterations that limit the self-association of RING1B, possibly helping to support a stable and enzymatically competent PRC1 complex. We have now included the quantification of these results in Figure 4—figure supplement 1 and described this interesting observation in the text (subsection “RYBP-dependent stimulation of PCGF1-PRC1 is associated with changes in the PCGF1-RING1B dimer but not with ubiquitin binding”). Importantly, however, the reduced level of PCGF1 does not explain activity differences between complexes, as complexes that contain RYBP simulate the relative activity of PRC1 to a much larger extent than can be explained by elevated PCGF1 levels (Figure 2) and both RYBP and YAF2 also significantly stimulate the activity of pre-existing RING1B/PCGF1 assemblies (Figure 2—figure supplement 1). Further structural work in the future will be imperative to understand the nature of this RYBP dependent stimulation.

*6) The Introduction, and actually most of the manuscript, does not give a clear overview of what has already been published on this topic. It is not that many major references were missing (although e.g. the work of the Sixma lab (Buchwald et al., 2006) should be included), rather the Introduction does not clearly summarized the work already done. In addition, somewhat more attention to work in the fly would be helpful to the reader as well. E.g. ncPRC1 complexes and there role in histone ubiquitylation were first described in the fly (Lagarou et al., 2008).*

We have now made extensive updates to the Introduction to cover in more detail the work already published in this area and we have specifically included observations from fly studies which we agree are important. Elsewhere in the manuscript, we have better summarized the published literature where appropriate.

*Finally, in the Results and Discussion sections, findings by others with similar experiments should be mentioned and where appropriated differences should be discussed. The whole manuscript strongly emphasizes the catalytic functions of PRC1/2 but this needs to be much better balanced by a discussion of the substantial evidence – both* in vitro *and* in vivo*, and in flies and mammals – that PRC1 executes many of its functions through a structural change in the underlying chromatin and that is independent of its catalytic activity.*

The reviewer(s) are correct that we have emphasized the catalytic activity of PRC1 and PRC2 as our study was focused on understanding the factors responsible for PRC1 catalytic activity in vitro (Figure 1) and how this relates to the acquisition of histone modifications in vivo. Unfortunately, from our current study we can infer very little about non-catalytic functions of PRC1, with the exception that gene expression changes appear to be more closely linked to reductions in the occupancy of PRC1/PRC2, not simply reductions H2AK119ub1/H3K27me3. We have now extensively altered the text and Discussion to indicate that PRC1 likely executes many of its functions through structural changes in the underlying chromatin structure independent of its catalytic activity.